# Delineating cooperative effects of Notch and biomechanical signals on patterned liver differentiation

Ishita Jain[1], Ian C. Berg[1], Ayusha Acharya[1], Maddie Blaauw[1], Nicholas Gosstola [1], Pablo Perez-Pinera[1] & Gregory H. Underhill [1✉]

Controlled in vitro multicellular culture systems with defined biophysical microenvironment have been used to elucidate the role of Notch signaling in the spatiotemporal regulation of stem and progenitor cell differentiation. In addition, computational models incorporating features of Notch ligand-receptor interactions have provided important insights into Notch pathway signaling dynamics. However, the mechanistic relationship between Notch-mediated intercellular signaling and cooperative microenvironmental cues is less clear. Here, liver progenitor cell differentiation patterning was used as a model to systematically evaluate the complex interplay of cellular mechanics and Notch signaling along with identifying combinatorial mechanisms guiding progenitor fate. We present an integrated approach that pairs a computational intercellular signaling model with defined microscale culture configurations provided within a cell microarray platform. Specifically, the cell microarray-based experiments were used to validate and optimize parameters of the intercellular Notch signaling model. This model incorporated the experimentally established multicellular dimensions of the cellular microarray domains, mechanical stress-related activation parameters, and distinct Notch receptor-ligand interactions based on the roles of the Notch ligands Jagged-1 and Delta-like-1. Overall, these studies demonstrate the spatial control of mechanotransduction-associated components, key growth factor and Notch signaling interactions, and point towards a possible role of E-Cadherin in translating intercellular mechanical gradients to downstream Notch signaling.

[1] Department of Bioengineering, University of Illinois at Urbana Champaign, Urbana, USA. ✉email: gunderhi@illinois.edu

During both tissue development and regeneration, multi-factor interactions between stem cells and their micro-environment lead to the segregation of functions by differentiation into various cell fates and the physical ordering of cell into complex patterns[1]. Studying these complex interactions of stem cells with their microenvironment becomes imperative to understand the basic processes of tissue development, congenital diseases, and injury responses. In liver development, bipotential progenitor cells surround the portal vein forming the ductal plate[2]. The progenitor cell layer directly in contact with the portal vein differentiates into cholangiocytes, the biliary epithelial cells, subsequently forming the bile ducts in the liver. Liver bipotential progenitor cells within the parenchyma, further away from the portal vein, differentiate into hepatocytes[2]. Numerous growth factors and signaling pathways have been shown to play a role in this multistep process of differentiation and morphogenesis. In particular, Wnt, HGF, and FGF primarily regulate hepatocytic differentiation[3–5], while Notch and TGFβ serve as critical regulators of biliary differentiation[6–8]. Abnormalities in the Notch signaling pathway have been implicated in various diseases such as the Alagille Syndrome, a developmental disorder and in repair/regeneration mechanisms in fatty liver disease[9]. In particular, mutations in the Notch ligand Jagged1 and Notch receptor NOTCH2 have been demonstrated to be the primary genetic abnormalities associated with Alagille syndrome[10–12], which leads to bile duct paucity and cholestasis. Utilizing high throughput microarray technology, we have previously reported on the effects of extracellular matrix composition on the differentiation of mouse embryonic liver progenitor cells[13,14]. Subsequently, we also found that TGFβ-induced biliary differentiation was significantly influenced by cooperative signals from substrate stiffness and extracellular matrix protein interactions[13]. These findings underscored the interplay of basic signaling mechanisms with the biomechanical cues in liver differentiation.

We have also examined patterned differentiation of bipotential liver progenitor cells when cultured within well-defined circular geometries[15]. In these studies, it was observed that the cells on the boundary of a circular multicell grouping differentiated toward the cholangiocytic lineage, whereas the cells on the interior, differentiated towards the hepatocytic lineage. Notch signaling, and cell traction forces, were required for this differentiation patterning. There have been several other systems reporting similar emergent patterned differentiation of stem cells as a result of the gradients in cellular forces. For example, Muncie et al. demonstrated gastrulation-like nodes associated with regions of high cell adhesive tension for embryonic stem cells cultured in constrained geometries[16]. Xue et al. demonstrated patterning of neuroectoderm tissue upon culturing the cells on circular geometry and observed similar gradients in cellular forces as in our system[17]. However, BMP signaling was demonstrated to be the main regulatory pathway controlling cell patterning in that system. Collectively, such studies are suggestive of the importance of intercellular biomechanical signals in guiding stem cell behavior, in combination with other soluble or cell–cell interaction pathways. Notably, cellular mechanics has been implicated in the cooperative regulation of other important developmental pathways such as Notch signaling. For example, mechanical forces have been implicated in Notch-receptor binding[18,19]. Hunter et al. showed that ligand–receptor binding interactions were not sufficient for maximal Notch signaling, and further, required actomyosin-based mechanical pulling forces[20]. In addition, Notch signaling modulation in response to vascular shear forces has been explored by Loerakker et al.[21]. Overall, although there have been numerous reports of mechanical force-induced alterations in Notch signaling, the detailed mechanism of how spatially regulated mechanical cues contribute to the differentiation patterning of stem and progenitor cells remains unclear.

Liver differentiation patterning within tightly controlled engineered microenvironments represents an excellent model system for examining the interactions between Notch signaling and cooperative microenvironmental cues. In these studies, we used this patterned differentiation system to develop a computational model-guided approach for analyzing the relationship between intercellular Notch signaling, multicellular geometry, and spatially regulated biomechanical cues. Specifically, we demonstrate the capability to integrate a computational model using ordinary differential equations for Notch signaling in cells within a circular lattice geometry. Experimentally determined traction force gradients and E-Cadherin functional patterns were additionally introduced into the model as a candidate regulator of Notch receptor–ligand interactions. Further, experimental validation and optimization of the microenvironment-responsive Notch signaling model were performed using cell microarray cultures and quantitative assessments of liver progenitor differentiation patterning. In particular, within the cell microarray cultures, the role of E-Cadherin and the Notch ligands, Jagged-1 (Jag1) and Delta-like-1 (Dll1) were experimentally evaluated using cell-intrinsic expression knockdown. Distinct differentiation patterning alterations were further observed in response to exogenous growth factor treatments in combination with the specific Notch ligand knockdowns and formed the basis for computational model adaptations that incorporated the relevant exogenous perturbation parameters consistent with the experimental findings.

## Results

**Patterned liver differentiation and underlying biomechanics and Notch signaling.** Cellular microarray technology was utilized to study liver differentiation which involved microcontact deposition-based printing of Collagen 1 protein circular domains (islands) in an array format on a polyacrylamide hydrogel on 12 mm coverslip. Bipotential mouse embryonic liver progenitor cells (BMELs) cultured on these 600 μm circular collagen I islands exhibited patterned differentiation. 10% of cells on the boundary ($0.95 \leq$ Radius $\leq 1$) were osteopontin (OPN) positive, which is a biliary marker, compared to less than 2% cells ($p$-value $< 0.001$) positive for OPN in the interior ($0.95 \leq$ Radius $\leq 1$). HNF4α, which is a hepatocyte-lineage transcription factor and marker, exhibited the opposite trend. Specifically, 20% of cells in the interior were HNF4α positive compared to less than 5% cells ($p$-value $< 0.05$) positive at the boundary (Fig. 1a–c). The bifurcation of the two cell fates occurred on the boundary where the number of HNF4α+ cells decreased exponentially and OPN+ cells increased exponentially. The thresholds to define the radius ranges for boundary ($0.95 \leq$ Radius $\leq 1$) and interior ($0 \leq$ Radius $\leq 0.95$) were decided based on this bifurcation of fate and used as the reference for all subsequent figures. In our previous reporting of this patterned differentiation, we had found that when Notch signaling was blocked using a γ-secretase inhibitor (GSI), a Notch pathway inhibitor, biliary differentiation was diminished[15]. Further, we also identified traction force gradients across the circular islands, where cells on the boundary exerted higher traction force (Fig. 1d, e). Similar traction force profiles have been previously reported both analytically, and also experimentally for other cell types[15,17,22,23]. Here, in order to delineate the complex interplay of Notch and biomechanics in patterned differentiation, we first aimed to directly quantify downstream Notch signaling activation using ISH-HCR (In Situ Hybridization-Hybridization Chain Reaction)[24] on the patterned microarrays. Hes1, a direct target of the Notch intracellular domain (NICD), has been shown to mediate biliary differentiation by increasing the expression of

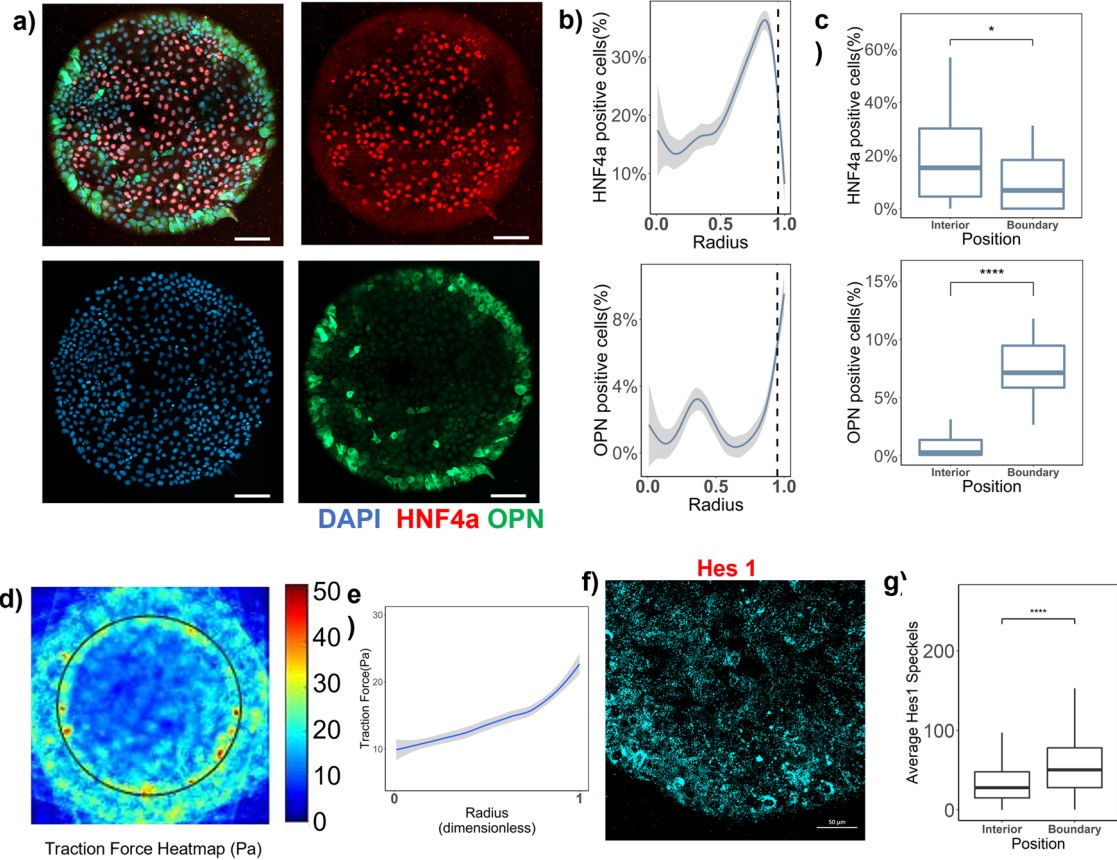

**Fig. 1 Patterned liver progenitor differentiation, cellular forces, and Notch signaling. a** Representative fluorescent Images of BMELs on Col 1 microarrays on 25 kPa polyacrylamide hydrogels. Red: HNF4a, Green: OPN, Blue: DAPI. Scale Bar: 100 μm. **b** Quantification of average percent HNF4a+ and OPN+ cells/island as a function of the radius. 0 is center, 1 is the edge, vertical dashed line at radius = 0.95. **c** Quantification average percent HNF4a+ and OPN+ cells/island in the interior (radius = 0–0.95) and at the boundary (radius= 0.95–1.00). **d** Traction force heatmap for BMELs on circular islands, blue (low traction) to red (high traction). **e** Traction force (Pa) of BMELs on circular microarrays as a function of the radius. **f** ISH-HCR quantification of Hes1 mRNA, 40X confocal maximum intensity projection of BMELs on circular islands. **g** Quantification of average Hes1 mRNA speckles on the boundary versus the interior. Boxplots: *p-value < 0.05; ****p-value < 0.0001, calculated using Wilcox test in R, Line plots—gray is the 95% confidence interval. n ≥ 4 biological replicates (independent experiments) and n ≥ 15 technical replicates (individual islands).

Sox9[8]. Hence, we measured Hes1 as a representative marker for Notch signaling on the circular islands. We found that the number of Hes1 mRNA speckles was significantly higher and exhibited a 2-fold increase (p-value < 0.0001) on the boundary compared to the interior of the circular microarrays (Fig. 1f, g). This increase on the boundary correlated with high traction force and biliary differentiation and was a direct confirmation of the role of Notch signaling in the patterned differentiation. Next, we wanted to elucidate how the cellular force gradients might be translating to Notch signaling gradients on the circular microarrays. Here, we report observed gradients in expression and localization of two proteins, E-Cadherin and filamentous actin across the islands (Supplementary Fig. 1). E-Cadherin expression was observed to be uniform in the cells on the interior and increased exponentially on the boundary. Filamentous actin was also constant across the circular microarray but decreased exponentially at the boundary. It was also qualitatively noted that the E-Cadherin seemed to localize to the cell membrane in the cells on the interior and was more diffused in the cells on the boundary. This was suggestive of a possible functional patterning of E-Cadherin influencing Notch signaling and resulting differentiation pattern.

**Design of computational model for understanding the patterned differentiation.** The next goal was to mechanistically

determine how various cell interaction and cellular forces were potentially regulating Notch signaling in the progenitor cells during differentiation. In order to understand the various factors involved in this patterned differentiation, a computational model was employed, to simulate Notch signaling in response to the complex interplay of cell-substrate forces and cell–cell interaction. The model was adopted from ref. [25], in which the interaction of the Notch receptor–ligand, how it leads to cleavage of the Notch Intracellular Domain (NICD), and feedback signaling to enhance/inhibit Notch receptor/ligand production, is modeled using ordinary differential equations based on the Collier Model[26]. This model when simulated in hexagonal-shaped cells in a regular lattice, resulted in a salt and pepper pattern, where a cell with higher Notch receptor expression is subsequently surrounded by the cells with higher ligand expression on all sides. Here, in addition to the Notch receptor–ligand interactions, the influence of biomechanical cues such as the traction force gradients, E-Cadherin expression, and localization gradients was integrated to systematically analyze the observed liver progenitor differentiation pattern. The model was also refined to be specifically introduce irregular polygon shapes on a circular lattice, which more closely recapitulate the individual cell shapes within the multicellular groupings defined by the microarray domains. Further, distinct functional responses for the two Notch ligands Dll1 and Jag1,

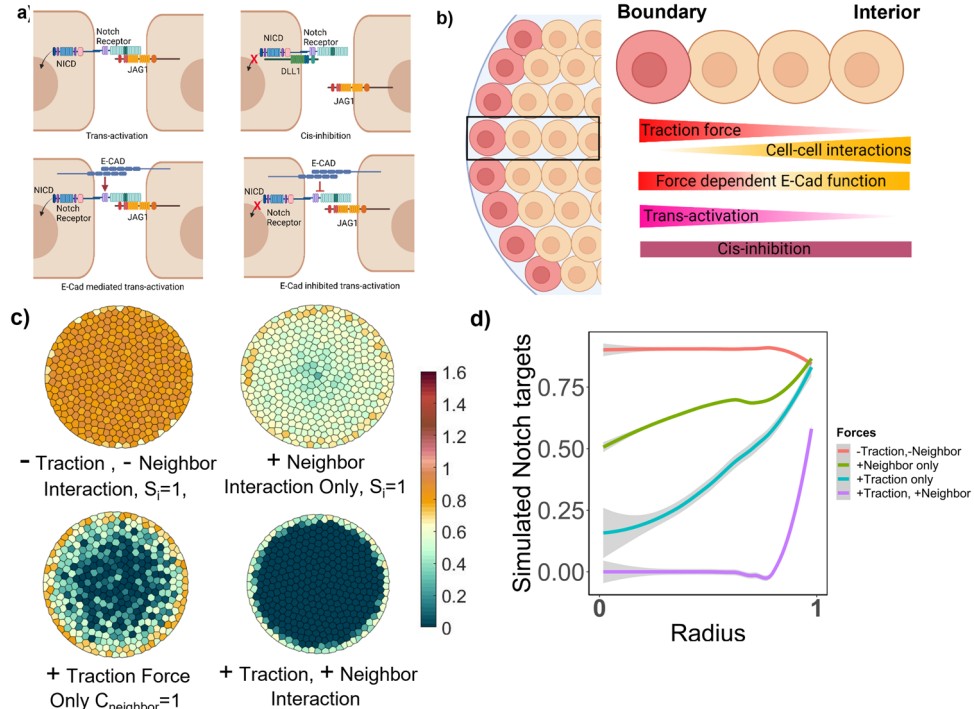

**Fig. 2 Computational modeling design for simulated Notch target genes in cells on circular geometry. a** Schematic of various Notch receptor–ligand interactions: trans-activation and cis-inhibition. Additionally, the bottom half of the schematic represents how E-Cadherin interacts with notch signaling. **b** Schematic of incorporation of the different Notch receptor–ligand interactions in cells on the boundary versus the interior. Made using biorender.com. **c** Simulated Notch target genes in cells on a circular lattice, with cellular forces gradients being added step by step. Heatmap color: arbitrary unit for Notch target genes. **d** Quantification of simulated Notch target genes in cells on a circular lattice for the different models in **c**. Gray: 95% confidence interval.

which exhibit differential relative trans-activation and cis-inhibition characteristics, was incorporated (Fig. 2a, b). In particular, it has been widely explored how the dynamics between different Notch receptor and ligands can lead to both activation and/or inhibition of downstream Notch targets, leading to patterns emerging during development[27–29]. Based on the experimental results from our previous work[13,15], Dll1 was modeled to mediate cis-inhibition primarily ($k_{cD} = 22$, $k_{tD} = 0.3$) and Jag1 was modeled to mediate only trans-activation ($k_{tJ} = 10$). On the

boundary of the circular microarrays, the cells experienced high traction forces and relatively lower cell–cell interactions, hence trans-activation was modeled to be mediated by traction forces. The cells in the interior have high cell–cell interactions combined with lower substrate traction forces, hence trans-activation was also modeled to be inhibited by the level of cell–cell interaction experienced by the respective individual cell.

Equations (1–5) show the ODEs for the concentration of notch receptor ($N_{Ri}$), notch ligand Dll1($D_i$,), notch ligand Jag1($J_i$), the

$$dN_{Ri} = \underbrace{\text{betaN} \cdot \left(\frac{NT_i^h}{1 + NT_i^h}\right)}_{\text{Expression term}} - \underbrace{k_{tD} \cdot N_{Ri} \cdot D_{\text{neighbour}} \cdot \left(\frac{S_i}{0.1 + C_{\text{neighbor}}}\right)}_{\text{Trans-activation by Dll1}} - \underbrace{k_{cD} \cdot N_{Ri} \cdot D_i}_{\text{Cis-inhibition by Dll1}} - \underbrace{k_{tJ} \cdot N_{Ri} \cdot J_{\text{neighbour}} \cdot \left(\frac{S_i}{0.1 + C_{\text{neighbor}}}\right)}_{\text{Trans-activation by Jag 1}} - \underbrace{N_i}_{\text{Decay Term}} \quad (1)$$

$$dNICD_i = \text{betaNICD} \underbrace{\left(\frac{k_{tD} \cdot N_{Ri} \cdot D_{\text{neighbour}} \cdot \left(\frac{S_i}{0.1 + C_{\text{neighbor}}}\right)}{1 + k_{tD} \cdot N_{Ri} \cdot D_{\text{neighbour}} \cdot \left(\frac{S_i}{0.1 + C_{\text{neighbor}}}\right)} + \frac{k_{tJ} \cdot N_{Ri} \cdot J_{\text{neighbour}} \cdot \left(\frac{S_i}{0.1 + C_{\text{neighbor}}}\right)}{1 + k_{tD} \cdot N_{Ri} \cdot D_{\text{neighbour}} \cdot \left(\frac{S_i}{0.1 + C_{\text{neighbor}}}\right)}\right)}_{\text{Cleavage of NICD with trans-activation mediated by Dll1 and Jag 1 ligand}} - \underbrace{\text{NICD}_i}_{\text{Decay Term}} \quad (2)$$

$$dD_i = \underbrace{\text{betaD} \cdot \left(\frac{1}{1 + NT_i^h}\right)}_{\text{Expression term}} - \underbrace{k_{tD} \cdot N_{R\text{neighbour}} \cdot D_i \cdot \left(\frac{S_i}{0.1 + C_{\text{neighbor}}}\right)}_{\text{Trans-activation by Dll1}} - \underbrace{k_{cD} \cdot N_{Ri} \cdot D_i}_{\text{Cis-inhibition by Dll1}} - \underbrace{D_i}_{\text{Decay Term}} \quad (3)$$

$$dJ_i = \underbrace{\text{betaJ}}_{\text{Expression term}} - \underbrace{k_{tJ} \cdot N_{R\text{neighbour}} \cdot J_i \cdot \left(\frac{S_i}{0.1 + C_{\text{neighbor}}}\right)}_{\text{Trans-activation by JAG1}} - \underbrace{J_i}_{\text{Decay term}} \quad (4)$$

$$dNT_i = \text{NICD}_i - NT_i \quad (5)$$

notch receptor intracellular domain (NICD$_i$), and notch target genes (NT$_i$) in each cell $i$ on the circular geometry. The equations in this set are based on a characteristic structure with several key terms: an expression term, interaction terms (for example trans-activation), and a natural decay term. In particular, betaJ, betaD and betaN are the expression rates for Jag1, Dll1 and Notch receptor. $k_{tD}$ and $k_{tJ}$ are the constants for the trans-activation interaction strength for Dll1 and Jag1 ligand, respectively. $K_{cD}$ is the constant for the cis-inhibition interaction strength for Dll1. $k_{tD}$, $k_{tJ}$, and $k_{cD}$ specifically only signify interaction between receptor and ligand. The result of the interaction in inhibiting/ mediating NICD cleavage is modeled in Eq. (2), where only trans-activation interactions lead to the subsequent cleavage and activation. Further, the interaction term includes neighbor interactions where the amount of ligand and receptor in the neighbor affects the notch signaling in cell i. A neighbor is defined as the cell $i$ sharing cell membrane interaction with another cell in the model. $N_{Rneighbour}$, $J_{neighbor}$ and $D_{neighbor}$ are the cumulative sum of the notch receptor, Jag1 and Dll1 in all the neighboring cells of the ith cells on the circular lattice. Additionally, all the interaction terms in the ODE incorporated the experimentally determined mechanics of the circular geometry using the two terms $S_i$ and $C_{neighbor}$. E-Cadherin was modeled to interact with Notch signaling across the whole island, where E-Cadherin had a contrasting function in a cell, depending on the traction force and cell–cell forces in that cell. This assumption was based on experimental results and various reports of E-Cadherin intersecting with the Notch signaling in different contexts[30,31]. Experimentally observed E-Cadherin gradients and normalized traction force patterns were incorporated in the computational models through the introduction of terms that influence Notch receptor–ligand binding and downstream expression of Notch targets. Specifically, S$_i$ is the term for the traction force affecting notch receptor–ligand binding and is modeled as a function of E-Cadherin expression gradient and traction force (Eq. 6). $C_{neighbor}$ is the term for cumulative cell–cell based forces in the neighbor of the cell i, affecting notch receptor–ligand binding and is modeled as a function of E-Cadherin concentration and cell–cell force where $j$ is the neighbor of the cell I (Eq. (7)). The cell–cell force was modeled as the inverse function of traction forces, which was constantly higher in the interior and decreased exponentially on the boundary. This assumption was based on literature[22,32] where force localization in multicellular monolayers was verified both experimentally and computationally. In these previous efforts, it was determined that in order for substrate traction forces to be localized on the edge of multicellular grouping, cells must pull on each other (i.e. exhibit cell–cell tension) within the interior region:

$$S_i = ECAD_i \cdot Traction_i \qquad (6)$$

$$C_{neighbor} = \sum ECAD_j \cdot CellCellForce_j \qquad (7)$$

Values of all parameters are listed and further described in Supplementary Table 1. Initial values of each of these parameters were chosen based on the code framework first established in ref. [25], which was also previously demonstrated in our early studies examining liver progenitor differentiation[15]. Starting from the parameter values used in these two papers, all parameters were varied across a range systematically, and the resulting simulated Notch target pattern was compared to the experimentally determined differentiation pattern. For example, two parameters (Notch receptor expression parameter (betaN) and NICD activation parameter (betaNICD)) were varied over a range to demonstrate how all the base model parameters were set, and these results are displayed in Supplementary Fig. 2. The initial condition for each cell had relatively low values of both Jag1 and Dll1, with some noise,

betaN levels of notch receptor, and 0 levels of NICD and notch target genes, adapted from[25]. The exact functions of these initial conditions could also be obtained using the getIC function in the MATLAB code and Supplemental Table 3. Subsequently, Eqs. (1–5) were simulated for 1000 iterations for 400 cells in a circular lattice.

Figure 2c, d shows the resulting simulated levels of the notch targets across the islands. The basic model with zero cell–cell forces, traction force, and E-Cadherin protein values is shown in the upper left corner of Fig. 2c. It matched the classic Collier Model[26], wherein a salt and pepper pattern is observed. However, as the influence of the cell–cell and traction forces were incorporated ($S_i$ and $C_{neighbor}$ terms integrated), a gradient in the pattern started to emerge. Finally, when both components were incorporated, a pattern similar to the experimentally measured OPN expression pattern emerged, in which substrate traction forces together with spatially dependent E-Cadherin function enhanced the gradient of active Notch signaling only on the edge, while cell–cell tension, modulated by a distinct action of E-Cadherin, inhibited active Notch signaling in the center. Subsequently, these computational model parameter values and assumptions encompassing the aggregated model were tested by experimentally perturbing various components and matching the computational Notch target expression pattern with the experimental OPN pattern.

**Contrasting effect of Notch ligand Dll1 and Jag1 knockdowns.** The functions of the two distinct progenitor-intrinsic ligands were verified experimentally by knocking down (KD) Dll1 and Jag1 in the BMEL cells using lentiviral transfection (Supp. Fig. 4). The OPN pattern is enhanced with Dll1 KD with a significant increase of OPN+ cells from 10% to 30% on the boundary. This can also be observed from the radial line graph of OPN positivity, which highlights that at $0.95 \leq Radius \leq 1$, a higher percentage of OPN+ cells were observed with Dll1 KD compared to the WT (Fig. 3a–c). Conversely, the percentage OPN+ cells significantly decreased from 10% to 0% following the Jag1 KD, as seen by both the line graph and box plot. These experimental results formed the basis for the two ligands exhibiting differential roles in the computational model for Notch signaling. It was also notable to observe the change in hepatocytic differentiation with the Notch ligand knockdown, which decreased significantly in response to the Dll1 KD, and modestly following the Jag1 KD. A slight decrease in average cell number/island was also observed with Dll1 KD and Jag1 KD, however, although the ligand knockdown cells maintained 100% confluent islands with uniform spatial distribution (Supplementary Fig. 3a, b). Traction force microscopy revealed significantly higher traction forces for the Dll1 KD cells at the boundary (radius = 1), whereas traction remained unchanged for the Jag1 KD compared to the control (Supplementary Fig. 3c, d). In silico knockdowns created with the computational model revealed a similar trend for the simulated Notch targets (Fig. 3d–f). In particular, when both the expression for Jag1 and Dll1 is reduced by 75% in the model (betaJ and BetaD reduced by 75%), we observed significantly diminished simulated Notch target gene expression pattern and an enhanced simulated Notch target gene expression pattern, respectively. The stability of the parameters betaJ and betaD in influencing the simulated Notch targets is demonstrated in Supplementary Fig. 8. It is important to note that even though Dll1 exhibited a majority cis-inhibitory role, its knockdown resulted in only enhancement of the biliary pattern on the boundary and does not result in ubiquitous OPN expression. This observation was suggestive of an additional cooperative signal that downregulates Notch signaling and biliary differentiation within the interior of the circular microarray cell domains.

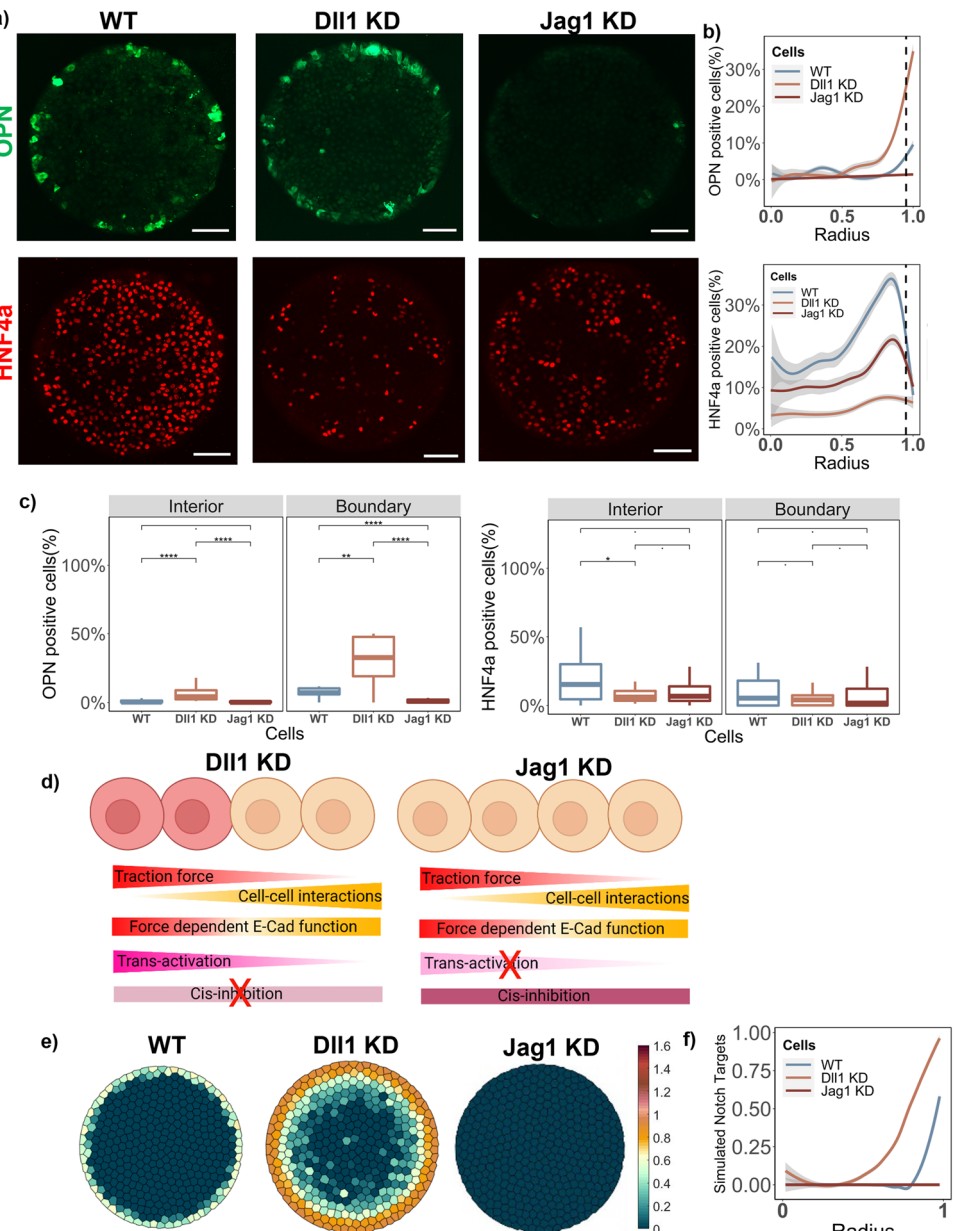

**Fig. 3 Effect of intrinsic Notch ligand knockdowns on patterned differentiation. a** Representative fluorescent Images of WT, Dll1 KD, and Jag1 KD BMELs on Col 1 microarrays on 25 kPa polyacrylamide hydrogels. Red: HNF4a, Green: OPN. Scale Bar: 100 μm. **b** Quantification of average percent HNF4a+ and OPN+ cells/island is as a function of the radius. 0 is center, 1 is the edge, vertical dashed line at radius = 0.95. **c** Quantification of average percent HNF4a + and OPN+ cells/island in the interior (radius = 0–0.95) and at the boundary (radius = 0.95–1.00). '.': ns; *p-value < 0.05; ***p-value < 0.001 ****p-value < 0.0001, calculated using Wilcox test in R. **d** Schematic of the simulated in silico knockdowns for Dll1 KD and Jag1 KD. Made using biorender.com. **e** Simulated Notch target genes on cells on a circular lattice, with in silico knockdown simulation. Heatmap color: arbitrary unit for Notch target genes. **f** Quantification of simulated Notch target genes in cells on a circular lattice for the in silico knockdowns. Line plots—Gray: 95% confidence interval. $n \geq 4$ biological replicates (independent experiments) and $n \geq 15$ technical replicates (individual islands).

**Downstream biomechanical signaling in patterned liver differentiation.** The underlying expression pattern of E-Cadherin (Supplementary Fig. 1) on the BMELs circular microarrays was an indicator of a potential role in influencing the liver bipotential progenitor differentiation pattern. We have also previously shown that when E-Cadherin is blocked in the BMEL cells using a function-blocking anti-E-cadherin antibody, DECMA-1, both cholangiocytic and hepatocytic differentiation was significantly diminished on the circular microarrays[15]. Here, to systematically investigate the contribution of E-Cadherin to progenitor Notch signaling and diffrentiation, E-Cadherin was knocked down using

lipid-based transfection of siRNA (ECad KD), and the reduction of E-Cadherin mRNA and protein expression was confirmed using RT-PCR (Supplementary Fig. 4). Following E-Cadherin knockdown, the primary observation was a decrease in the biliary differentiation on the boundary and complete loss of hepatocytic differentiation (Fig. 4a–c). Notably, the traction force gradient in ECad KD cells was maintained with higher traction force across the whole island compared to the negative control (Fig. 4d, e). Since the traction force gradient observed in BMELs on circular islands is a collective phenomenon of balancing local cell–cell forces and contractile force on the surface, it was hypothesized

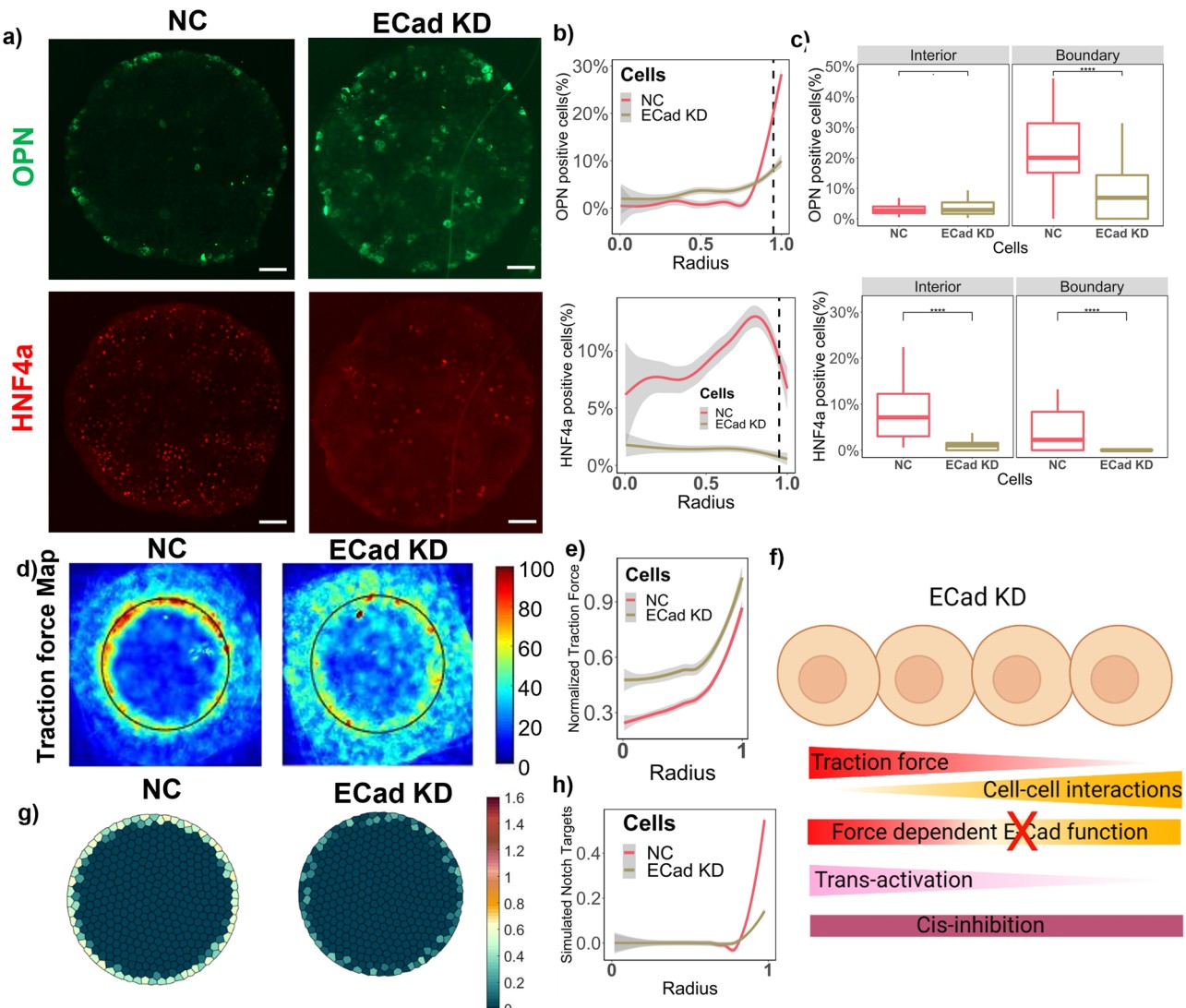

**Fig. 4 Effect of intrinsic E-Cadherin siRNA knockdown on patterned differentiation. a** Representative fluorescent Images of Negative Control (NC), and ECad KD BMELs on Col 1 microarrays on 25 kPa polyacrylamide hydrogels. Red: HNF4a, Green: OPN. Scale Bar: 100 µm. **b** Quantification of average percent HNF4a+ and OPN+ cells/island as a function of the radius. 0 is center, 1 is the edge, vertical dashed line at radius = 0.95. **c** Quantification of average percent HNF4a+ and OPN+ cells/island in the interior (radius = 0–0.95) and at the boundary (radius = 0.95–1.00). '.': ns; ****$p$-value < 0.0001, calculated using Wilcox test in R. **d** Traction force heatmap for NC and E-Cad KD BMELs on circular islands, blue (low traction) to red (high traction). **e** Traction force (kPa) of BMELs as a function of the radius, 0 being the center and 1 being the edge. **f** Schematic of the simulated in silico knockdown for E-Cadherin. Made using biorender.com. **g** Simulated Notch target genes on cells on a circular lattice, with in silico knockdown simulation. Heatmap color: arbitrary unit for Notch target genes. **h** Quantification of simulated Notch target genes in cells on a circular lattice for the in silico knockdowns. Line plots—Gray: 95% confidence interval. $n \geq 4$ biological replicates (independent experiments) and $n \geq 15$ technical replicates (individual islands).

that complete cell–cell force disruption would have resulted in uniform traction force across the whole island[22,32]. Consequently, the maintenance of a traction force gradient in the ECad KD cells is suggestive of intact cell–cell forces despite E-Cadherin loss. These cell–cell forces could have possibly been mediated by other cadherins, and hence retaining the cellular disc functionality. Furthermore, the alterations in differentiation patterning indicated a role of E-Cadherin downstream of the cellular mechanics and a possible role in influencing Notch signaling directly. Since the cellular forces on the E-Cadherin adherens junctions and associated cytoskeletal elements may be different in the distinct regions of the multicellular island (i.e., boundary versus the interior), the nature of interaction of E-Cadherin with the Notch signaling may also be spatially dependent. Based on this possibility, we adapted the model to incorporate a spatially dependent force response behavior of E-cadherin. Specifically, in the model,

E-cadherin facilitated trans-activation of the Notch receptor as a function of the traction force and inhibited trans-activation as a function of the cell–cell forces on the island. For the simulation of the ECad KD in the computational model, E-Cadherin expression was reduced by 70% (simulated 70% knockdown) uniformly across the cell island, which caused a decrease in the simulated Notch targets (Fig. 4f–h). The stability of the ECAD parameter in influencing the simulated Notch target values is demonstrated in Supplementary Fig. 8. It was only when such a force-dependent function of E-Cadherin was incorporated in the computational model, all subsequent perturbations aligned with experimental results. Interestingly, there was also a significant decrease in the average of number of cells/island observed with ECad KD (Supplementary Fig. 5). Overall, the final aggregate model outlined in Fig. 2 was the result of multiple reiteration cycles and parameter optimization, with experimental verification of each addition.

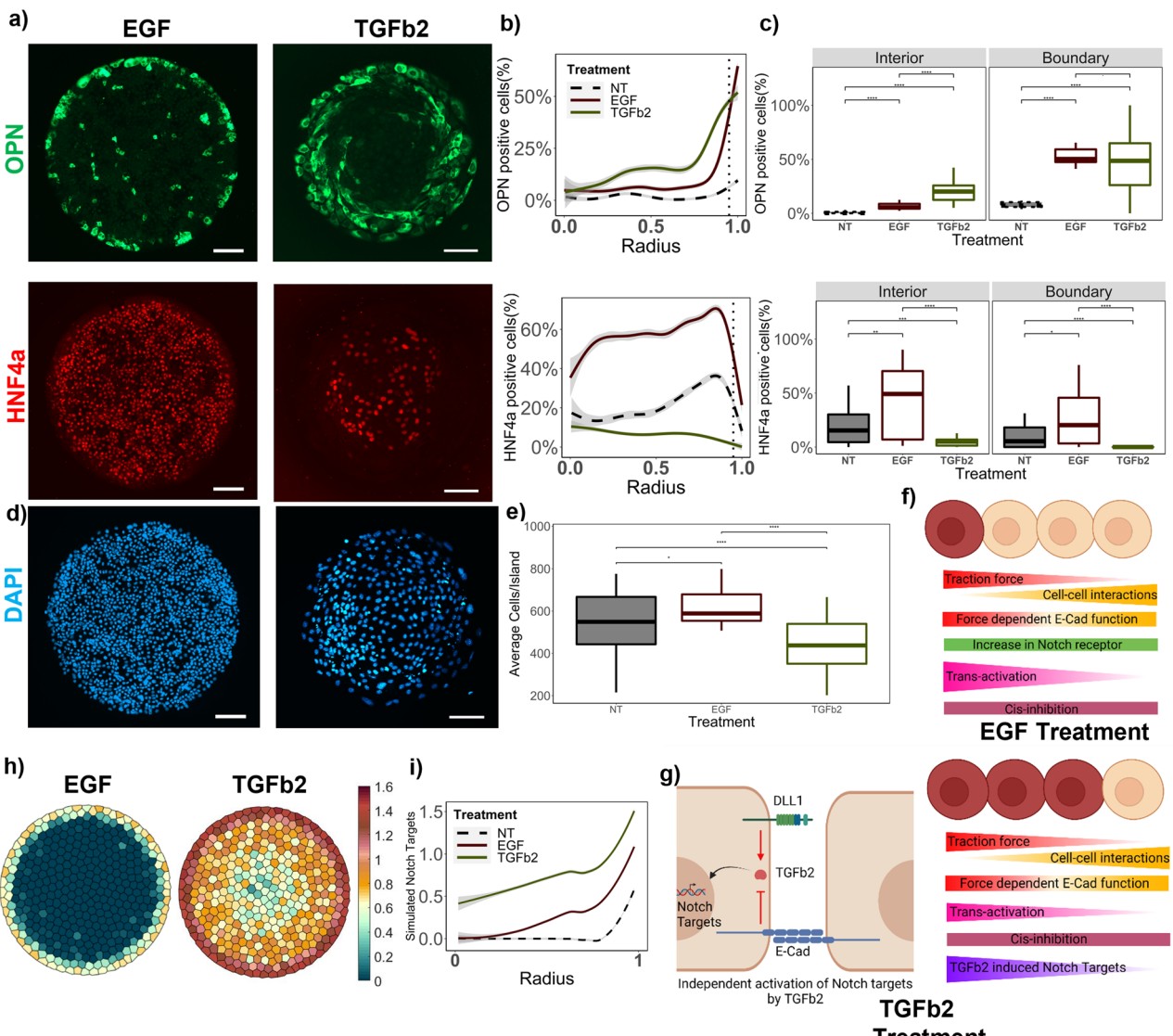

**Fig. 5 Effect of extrinsic growth factor treatments on patterned differentiation. a** Representative fluorescent Images of EGF and TGFβ2 treated wild-type BMELs on Col 1 microarrays on 25 kPa polyacrylamide hydrogels. Red: HNF4a, Green: OPN, Blue: DAPI. Scale Bar: 100 μm. **b** Quantification of average percent HNF4a+ and OPN+ cells/island as a function of the radius. 0 is center, 1 is the edge, vertical dashed line at radius = 0.95. **c** Quantification of average percent HNF4a+ and OPN+ cells/island in the interior (radius = 0–0.95) and at the boundary (radius = 0.95–1.00). **d** DAPI fluorescent Images for EGF and TGFβ2 treated wild-type BMELs on circular microarrays. Scale Bar: 100 μm. **e** Quantification of Average Cell Number per individual island for EGF and TGFβ2 treated wild-type BMELs. **f, g** Schematic of the simulated in silico growth factor treatments in the base model. Made using biorender.com. **h** Simulated Notch target genes on cells on a circular lattice, with in silico growth factor treatments. Heatmap color: arbitrary unit for Notch target genes. **i** Quantification of simulated Notch target genes in cells on a circular lattice for the in silico growth factor treatments. Line plots—Gray: 95% confidence interval. Boxplots—'.': ns; *p-value < 0.05; **p-value < 0.01; ***p-value < 0.001 ****p-value < 0.0001, calculated using Wilcox test in R. $n \geq 4$ biological replicates (independent experiments) and $n \geq 15$ technical replicates (individual islands). The No Treatment Control (NT) for the wild-type BMELs is referred from Fig. 3b, c for statistical comparison.

**Extrinsic perturbations with growth factors.** As a consequence of the various Notch and cell–cell adhesion pathway knockdown alterations, we perturbed Notch signaling and differentiation intrinsically, thus providing insights into progenitor-intrinsic factors contributing to Notch signaling and differentiation patterning. Subsequently, we sought to investigate the effect of extrinsic signals, such as growth factor treatments, to act as cooperative regulators, thereby recapitulating aspects of paracrine signaling in vivo. In particular, we examined the effect of epidermal growth factor (EGF) based on its reported interaction with Notch signaling during liver development[33,34]. The first observation following EGF treatment of BMEL cells within cell microarrays was the increase in both hepatocytic and biliary differentiation. Specifically, there was an increase

from 10% to 60% percent OPN+ cells on the boundary and 20% to 50% HNF4α+ cells in the interior. Furthermore, this increase occurred together with a retention of the differentiation pattern (Fig. 5a–c). The average cell number/island increased significantly with EGF treatment, which is suggestive of a potential increase in proliferation (Fig. 5d, e). However, the fold-increase of OPN+ and HNF4α+ cells was at least 4-fold higher than the increase in the number of cells. To computationally model EGF treatment, upregulation of Notch receptor expression was applied to the aggregate model by adding a term $egf*N_{receptor}$ in the expression function, where egf is the parameter for the EGF effect. (Fig. 5f). The implementation of this straightforward perturbation, based on previous reports of increase in Notch1 receptor expression with EGF

treatment[33], was effective to emulate the experimental findings. A definitive increase in the simulated Notch target levels in the wild type cells, with retention of spatial patterning, was observed. The stability of the egf parameter in increasing Notch receptor production and resulting simulated notch targets is demonstrated in Supplementary Fig. 8:

$$dN_{Ri} = \underbrace{\text{betaN} \cdot \left( \frac{NT_i^h + egf \cdot N_{Ri}}{1 + NT_i^h + egf \cdot N_{Ri}} \right)}_{\text{Expression term}} - \underbrace{k_{tD} \cdot N_{Ri} \cdot D_{\text{neighbour}} \cdot \left( \frac{S_i}{0.1 + C_{\text{neighbor}}} \right)}_{\text{Trans-activation by DLL1}}$$
$$- \underbrace{k_{cD} \cdot N_{Ri} \cdot D_i}_{\text{Cis-inhibition by DLL1}} - \underbrace{k_{tJ} \cdot N_{Ri} \cdot J_{\text{neighbour}} \cdot \left( \frac{S_i}{0.1 + C_{\text{neighbor}}} \right)}_{\text{Trans-activation by JAG1}} - \underbrace{N_i}_{\text{Decay term}} \quad (8)$$

Transforming growth factor β (TGFβ) signaling is one of the most important signaling pathways during biliary development and its interaction with Notch signaling is studied often[35,36]. We have extensively studied the role of TGFβ1 on biliary differentiation and Notch signaling in BMEL cells. Here, we aimed to explore the effect of TGFβ2 as another dimension of perturbation, which has also been implicated to influence biliary differentiation and is specifically localized in the periportal region during liver bud development[37]. Further, in previous studies, we observed that the treatment of BMEL cells within circular microarray domains with 1.5ng/mL concentration of TGFβ1 served to override the mechanical influence of the circular geometry, resulting in OPN+ cells across the whole island[15]. However, here we demonstrate that the introduction of TGFβ2, at a relatively lower overall media concentration 0.1 ng/mL, resulted in a 6-fold increase in the percentage of OPN+ cells within the boundary region. Further, we also observed the OPN pattern encroaching inwards, implied by an exponential increase of OPN+ cells at lower radius ~0.75 compared to radius~0.9 for control condition (Fig. 5a–c). Lastly, a change in morphology of the OPN+ cells was observed, with the cells exhibited a more elongated shape and some spiraling behavior. A decrease in the average cell number/island was observed with the TGFβ2 treatment compared to the No Treatment control (Fig. 5d, e). For the simulation of TGFβ2 effects, multiple iterations to the model were required and experimental data from the various knockouts were imperative to the design of a base treatment modification in the model. There have been previous reports of TGFβ2 upregulating Notch ligands as well as upregulating the Notch target gene expression independent of canonical Notch signaling[38,39]. Hence, three modifications were made to simulate TGFβ2 signaling in silico (Fig. 5g): 1) Upregulation of DLL1 ligand (Eq. 9); 2) Upregulation of JAG1 ligand (Eq. 10); 3) Addition of a new interaction term between DLL1 ligand and TGFβ2, influenced by cellular forces, upregulating Notch target genes directly (Eq. 11). Finally, the in silico TGFβ2 treatment resulted in similar changes in the pattern of the Notch target genes as we see for experimentally measured OPN (Fig. 5h–i). The stability of the tgf parameter in increasing Notch target values is demonstrated in Supplementary Fig. 8:

$$dJ_i = \underbrace{\text{betaJ} \cdot \left( 1 + \frac{tgfJ \cdot J_i}{1 + tgfJ \cdot J_i} \right)}_{\text{Expression term}}$$
$$- \underbrace{k_{tJ} \cdot N_{\text{Rneighbour}} \cdot J_i \cdot \left( \frac{S_i}{0.1 + C_{\text{neighbor}}} \right)}_{\text{Trans-activation by JAG1}} - \underbrace{J_i}_{\text{Decay term}} \quad (9)$$

$$dD_i = \text{betaD} \cdot \underbrace{\left( \frac{1 + tgfD \cdot D_i}{1 + NT_i^h + tgfD \cdot D_i} \right)}_{\text{Expression term}}$$
$$- \underbrace{k_{tD} \cdot N_{\text{Rneighbour}} \cdot D_i \cdot \left( \frac{S_i}{0.1 + C_{\text{neighbor}}} \right)}_{\text{Trans-activation by DLL1}} - \underbrace{k_{cD} \cdot N_{Ri} \cdot D_i}_{\text{Cis-inhibition by DLL1}} - \underbrace{D_i}_{\text{Decay term}} \quad (10)$$

$$dNT_i = \text{NICD}_i + \underbrace{tgf \cdot D_i^b \cdot \left( \frac{1}{0.1 + \nu \cdot C_{\text{neighbor}}^d} \right)}_{\text{TGFb induced expression of Notch Targets}} - \underbrace{NT_i}_{\text{Decay term}} \quad (11)$$

**Notch ligand knockdowns with growth factor treatments.** Combining both intrinsic and extrinsic perturbations further provided insights into the cooperative effects underlying the liver progenitor differentiation pattern. For the combined Dll1 KD cells and EGF treatment condition, we noted a dramatic increase in the percent OPN+ cells in the interior compared to Dll1 KD cells with no growth factor treatment (Fig. 3a–c), which rises from 0 to 35%, in addition to a 6-fold increase at the boundary (Fig. 6a–c). A decrease in the percent HNF4α+ cells for Dll1 KD cells with no growth factor treatment compared to the WT was observed (Fig. 3a–c), and that further decreased to 0 with EGF treatment on the same cells (Supplementary Fig. 6). Furthermore, a 1.5 fold increase (p-value < 0.0001) in the number of Hes1 speckles was observed in the interior with Dll1 KD cells treated with EGF compared to WT cells treated with EGF, collectively demonstrating higher active notch signaling in the interior (Fig. 6h, i). A contrasting result is observed with TGFβ2 treatment on the Dll1 KD cells, where only a slight increase in the percent OPN+ cells is observed (Fig. 6a–c) compared to 6-fold-increase in the WT cells with TGFβ2 treatment (Fig. 5a–c). Furthermore, the decrease in the percent HNF4α+ cells observed with EGF treatment is reversed with TGFβ2 treatment for the Dll1 KD cells (Supplementary Fig. 6). These findings prompted the hypothesis that TGFβ2 may interact with the Dll1 ligand in a manner to independently upregulate Notch target genes in the computational model. Accordingly, following Dll1 KD, the TGFβ2 treatment effect on the biliary differentiation is significantly reduced. In contrast, a substantial increase in the levels of simulated Notch target was observed with EGF simulation specially in the region r = 0–0.75 whereas only a slight increase is observed with the TGFβ2 simulation (Fig. 6d, g).

For the Jag1 KD cells with exogenous EGF treatment, we noted a slight yet significant increase of 25% in OPN+ cells on the boundary and a slight decrease in the number of HNF4a+ cells all across the island (Fig. 7a–c, Supplementary Fig. 6) compared to Jag1 KD cells with no growth factor treatment (Fig. 3a–c). However, following TGFβ2 treatment, we observed an increase in the percent OPN+ cells even in the interior of the islands for Jag1 KD cells. A 10-fold increase in percent OPN+ cells is observed on the interior with the TGFβ2 treatment compared to no treatment in the JAG1 KD cells (Fig. 7a–c). Additionally, the loss of the boundary-restricted Notch signaling and differentiation pattern was supported by Hes1 quantification. Specifically, the number of Hes1 speckles exhibited a 1.9 fold increase (p-value < 0.0001) in the interior in JAG1 KD cells treated with TGFβ2 compared to non-treated JAG1 KD cells (Fig. 6h, i). For both the Dll1 KD and Jag1 KD, there was an increase in average cell number/ island with EGF treatment and a decrease after TGFβ2 treatment

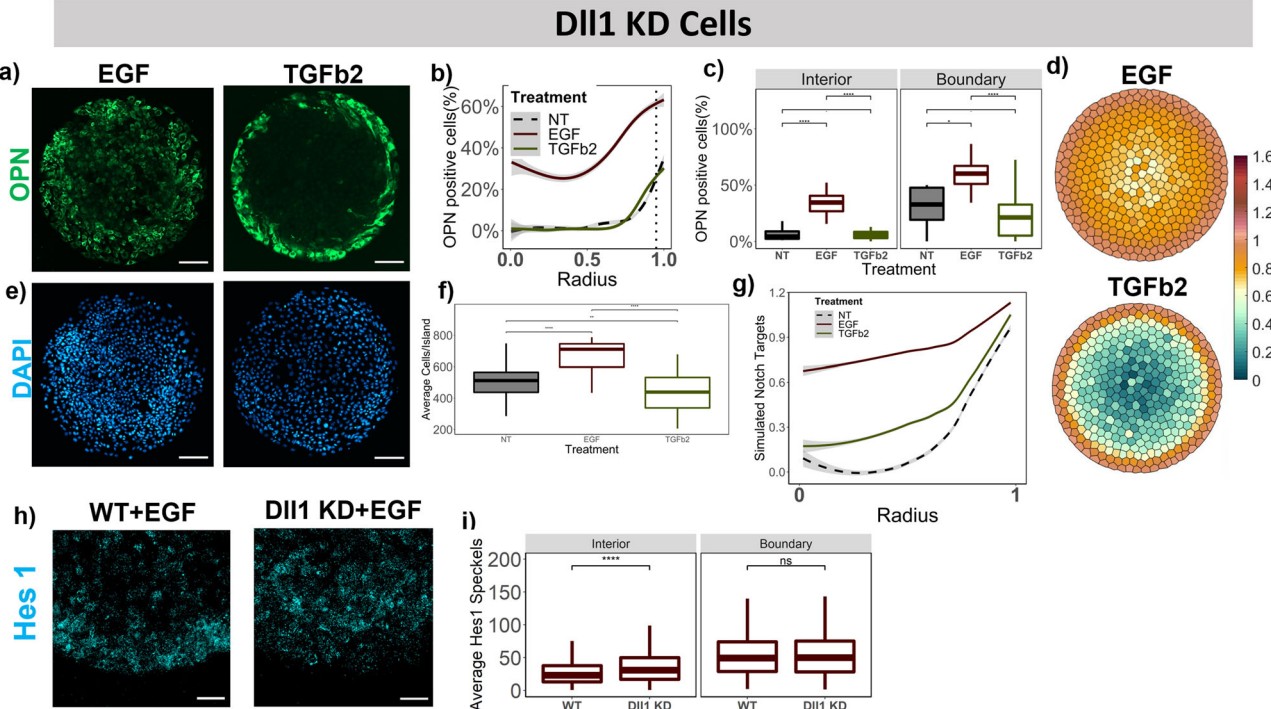

**Fig. 6 Effect of extrinsic growth factor treatments in conjunction with Dll1 knockdowns. a** Representative fluorescent Images of EGF and TGFβ2 treated Dll1 KD BMELs on Col 1 microarrays on 25 kPa polyacrylamide hydrogels. Green: OPN, Blue: DAPI. Scale Bar: 100 μm. **b** Quantification of average percent OPN+ cells/island as a function of the radius. 0 is center, 1 is the edge, vertical dashed line at radius = 0.95. **c** Quantification of average percent OPN+ cells/island in the interior (radius = 0–0.95) and at the boundary (radius = 0.95–1.00). **e** DAPI fluorescent Images for EGF and TGFβ2 treated Dll1 BMELs on circular microarrays. Scale Bar: 100 μm. **f** Quantification of Average Cell Number per individual island for EGF and TGFβ2 treated Dll1 KD BMELs. **d** Simulated Notch target genes on cells on a circular lattice, with in silico growth factor treatments and Dll1 KD. Heatmap color: arbitrary unit for Notch target genes. **g** Quantification of simulated Notch target genes in cells on a circular lattice for the in silico growth factor treatments and Dll1 KD. **h** ISH-HCR quantification of Hes1 mRNA, 40X confocal maximum intensity projection of wild-type and Dll1 KD BMELs treated with EGF on circular islands. **i** Quantification of average Hes1 mRNA speckles for images in 6n on the boundary versus the interior. Boxplots- '.': ns; *p-value < 0.05; **p-value < 0.01; ***p-value < 0.001; ****p-value < 0.0001, calculated using Wilcox test in R. Line plots—Gray: 95% confidence interval. $n \geq 4$ biological replicates (independent experiments) and $n \geq 15$ technical replicates (individual islands). The No Treatment Control (NT) for the Dll1 KD BMELs is referred from Fig. 3b, c for statistical comparison.

(Figs. 6e, f and 7e, f). The model simulation of JAG1 KD (betaJ reduced by 75%) and growth factor treatments (Eqs. (8–11)), resulted in a similar increase in simulated Notch targets with EGF, where the increase was limited to the boundary and drastic increase with TGFβ2 simulations, where the increase was across the whole island (Fig. 7d, g).

**E-Cadherin knockdown with extrinsic growth factor treatments.** The E-Cadherin knockdown together with EGF treatment resulted in a similar increase in the OPN pattern as with the WT cells; however, the restriction of the OPN+ cells to the boundary was reduced, with an increased percent OPN+ cells both in the interior and boundary following EGF treatment (Fig. 8a–c). A similar increase was observed in the simulated Notch targets with the simulated ECad knockdown (E-Cadherin expression reduced by 70% uniformly across entire island) together with the EGF treatment model (Fig. 5h, Eq. (8)). Furthermore, TGFβ2 treatment resulted in a uniform increase in the percent OPN+ cells across the whole island (Fig. 8a–c). This was confirmed by a vertical shift on the line graph in Fig. 8b and a 10-fold increase in the interior in Fig. 8c. This led us to hypothesize a role of E-Cadherin and cellular forces in TGFβ2 supplementing Notch signaling and was then incorporated in the adapted TGFβ2 treatment (Fig. 5f, Eq. (11)) model. Upon simulating Notch targets within model following simulated TGFβ2 treatment, an island-wide increase in Notch signaling was observed similar to the experimental OPN

quantification (Fig. 8d, g). Further, Hes1 mRNA quantification demonstrated an increased expression in the interior for ECad KD cells treated with TGFβ2 compared to non-treated ECad KD cells (Fig. 8h, i). The cell number/island showed a similar trend of an increase with EGF treatment and a decrease with TGFβ2, similar to the other experiments with WT, Dll1 KD and Jag1 KD BMEL cells. Lastly, following E-Cadherin knockdown, HNF4α expression was increased at both the boundary and interior following EGF treatment and was not affected by TGFβ2 treatment (Supplementary Fig. 7).

**Discussion**
We implemented a combined experimental and computational approach to systematically evaluate multicellular cellular fate patterning as a function of biophysical forces and Notch signaling in the context of liver progenitor differentiation. Our previous studies reporting the patterned differentiation of bipotential liver progenitor cells within a 2D circular geometry paved the way to study the complex mechanism of the interplay between Notch signaling and cellular biomechanics in cellular fate patterning. Unique patterns in the underlying cell–cell signaling and traction force distributions were identified and quantitatively assessed. It was observed that the cells exert a higher traction force on the boundary, correlating with relatively increased biliary differentiation in this perimeter region. To measure the spatial expression of a direct Notch target gene, Hes1, a fluorescent

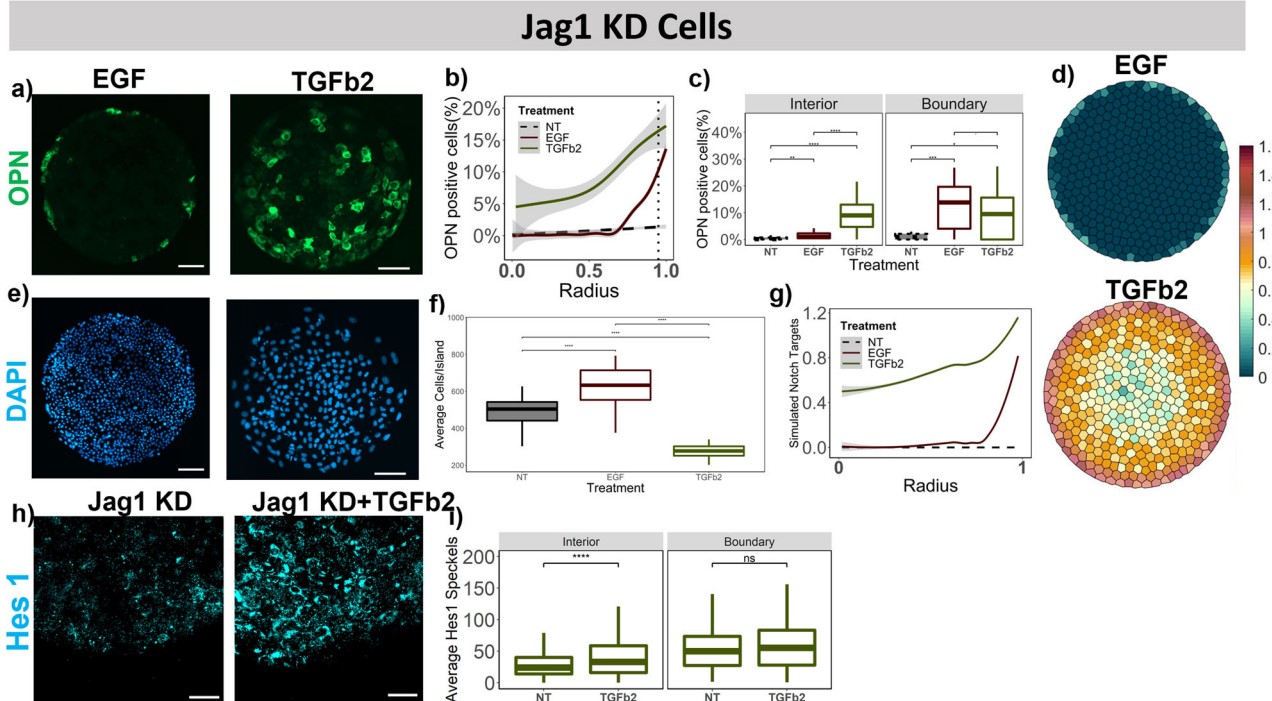

**Fig. 7 Effect of extrinsic growth factor treatments in conjunction with Jag1 knockdowns. a** Representative fluorescent Images of EGF and TGFβ2 treated Jag1 KD BMELS on Col 1 microarrays on 25 kPa polyacrylamide hydrogels. Green: OPN, Blue: DAPI. Scale Bar: 100 μm. **b** Quantification of average percent OPN+ cells/island as a function of the radius. 0 is center, 1 is the edge, vertical dashed line at radius = 0.95. **c** Quantification of average percent OPN+ cells/island in the interior (radius = 0–0.95) and at the boundary (radius = 0.95–1.00). **e** DAPI fluorescent Images for EGF and TGFβ2 treated Jag1 BMELs on circular microarrays. Scale Bar: 100 μm. **f** Quantification of Average Cell Number per individual island for EGF and TGFβ2 treated Jag1 KD BMELs. **d** Simulated Notch target genes on cells on a circular lattice, with in silico growth factor treatments and Jag1 KD. Heatmap color: arbitrary unit for Notch target genes. **g** Quantification of simulated Notch target genes in cells on a circular lattice for the in silico growth factor treatments and Jag1 KD. **h** ISH-HCR quantification of Hes1 mRNA, 40X confocal maximum intensity projection of Jag1 KD BMELs with and without TGFβ2 on circular islands. **i** Quantification of average Hes1 mRNA speckles for images in 6p on the boundary versus the interior. Boxplots- '.': ns; *p-value < 0.05; **p-value < 0.01; ***p-value < 0.001 ****p-value < 0.0001, calculated using Wilcox test in R. Line plots—Gray: 95% confidence interval. n ≥ 4 biological replicates (independent experiments) and n ≥ 15 technical replicates (individual islands). The No Treatment Control (NT) for the Jag1 KD BMELs is referred from Fig. 3b, c for statistical comparison.

in situ hybridization (FISH) technique (ISH-HCR) was utilized[24]. Even though the technique ISH-HCR uses secondary amplification of the mRNA FISH signal, the relative signal-to-noise ratio was comparatively low compared to immunofluorescence. However, we still observed an increased localization of Hes1 mRNA speckles on the boundary compared to the interior. Overall, this observation is consistent with the role of patterned Notch signaling in the resultant patterning of biliary versus hepatocyte differentiation in this in vitro differentiation system. Further, we observed higher E-Cadherin expression on the boundary, although this E-Cadherin expression was more diffused compared to the interior. Initially, this observation was counterintuitive as we initially anticipated elevated E-Cadherin within the interior of the islands, where each cell is surrounded by another cell on all sides and likely experiencing uniform adherensjunction mechanical tension. This shift towards more diffused E-Cadherin at the boundary could be indicative of increased turnover of the protein, which could be a function of the substrate traction forces[40]. The observed exponential increase in E-Cadherin expression at the perimeter was complemented by the patterned distribution of the filamentous actin. Filamentous actin was uniformly higher on the interior and decreased exponentially at the same radial location where E-Cadherin increased in expression. There have been reports of different morphological actin configurations, namely a cortical ring structure versus stress fibers in response to mechanical cues[41]. Collectively, the

interrelated shifts in expression and cellular localization of E-cadherin and F-actin across the cell domains are suggestive of a role of E-cadherin in actomyosin mechanical signaling as part of patterned differentiation response.

To systematically evaluate the function of the various components in the resulting differentiation pattern, we developed an ordinary differential equation-based computational model to compliment the experimental results. The model was aimed at simulating Notch signaling, with various biomechanical cues integrated into the model framework. The primary biomechanical parameters that were incorporated included the observed E-Cadherin expression pattern and measured cellular traction forces. Based on a series of reports of E-Cadherin mediating interaction of Notch receptors and ligand[30,31], we modeled the specific interaction of E-Cadherin with the Notch receptor influenced by the patterned expression and qualitative localization of the E-cadherin protein and incorporated a cellular forcedependent function of E-Cadherin. This model design was also motivated by the loss of both hepatocytic and biliary differentiation with E-Cadherin knockdown in progenitor cells. Additionally, there have been multiple previous investigations of force localization in circular multicellular monolayers such as in refs. [22,32], where they report similar traction force gradients as our model. These studies also report that the traction force gradients observed are the result of increased cell–cell tension on the interior, as part of a force balance mechanism that aims to

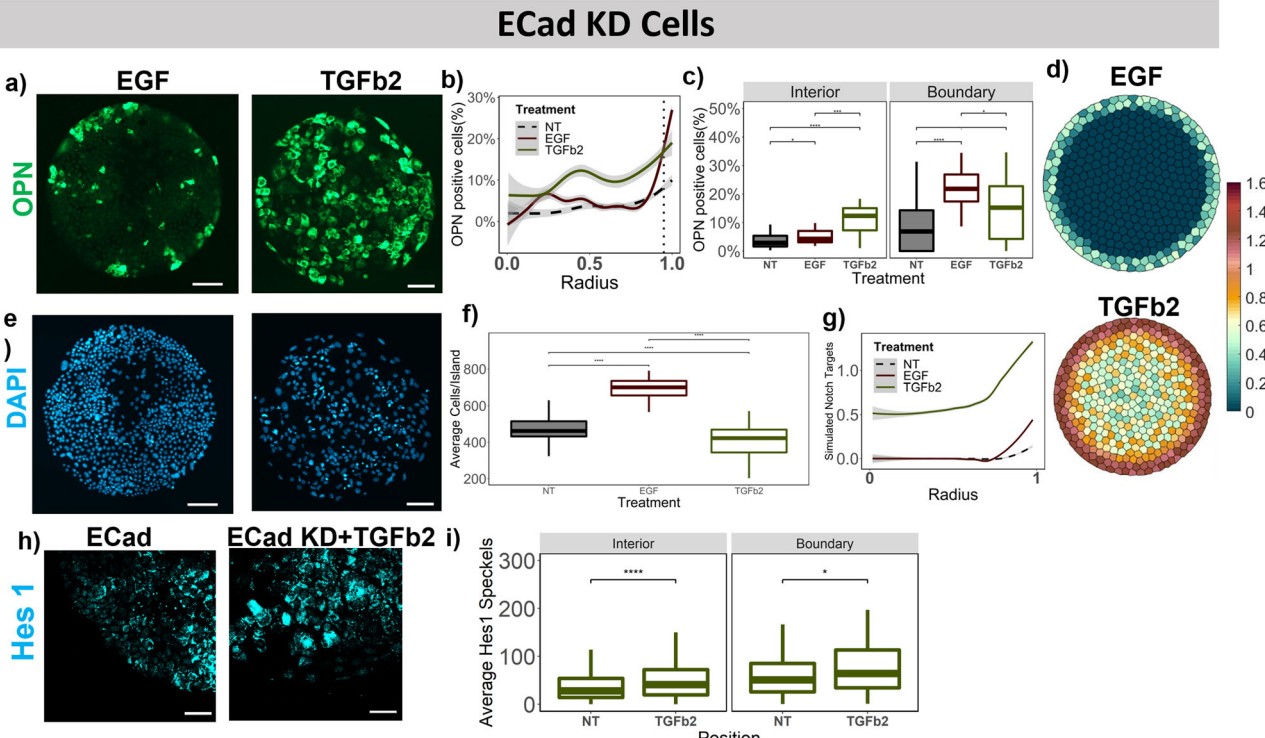

**Fig. 8 Effect of extrinsic growth factor treatments in conjunction with E-Cadherin siRNA knockdown. a** Representative fluorescent Images of EGF and TGFβ2 treated ECad KD BMELs on Col 1 microarrays on 25 kPa polyacrylamide hydrogels. Green: OPN, Blue: DAPI. Scale Bar: 100 μm. **b** Quantification of average percent OPN+ cells/island as a function of the radius. 0 is center, 1 is the edge, vertical dashed line at radius = 0.95. **c** Quantification of average percent OPN+ cells/island in the interior (radius = 0–0.95) and at the boundary (radius = 0.95–1.00). **e** DAPI fluorescent Images for EGF and TGFβ2 treated ECad KD BMELs on circular microarrays. Scale Bar: 100 μm. **f** Quantification of Average Cell Number per individual island for EGF and TGFβ2 treated ECad KD BMELs. **d** Simulated Notch target genes on cells on a circular lattice, with in silico growth factor treatments and E-Cadherin knockdown. Heatmap color: arbitrary unit for Notch target genes. **g** Quantification of simulated Notch target genes in cells on a circular lattice for the in silico growth factor treatments and ECad KD. **h** ISH-HCR quantification of Hes1 mRNA, 40X confocal maximum intensity projection of ECad KD BMELs with and without TGFβ2 on circular islands. **i** Quantification of average Hes1 mRNA speckles for images in 7h on the boundary versus the interior. Boxplots—'.': ns; *p-value < 0.05; **p-value < 0.01; ***p-value < 0.001 ****p-value < 0.0001, calculated using Wilcox test in R. Line plots—Gray: 95% confidence interval. n ≥ 4 biological replicates (independent experiments) and n ≥ 15 technical replicates (individual islands). The No Treatment Control (NT) for the Ecad KD BMELs is referred from Fig. 4b, c for statistical comparison.

balance forces between the interior and boundary regions. Hence, E-Cadherin mediated cell–cell forces (high in the interior) were modeled to inhibit trans-activation of the Notch receptor and E-Cadherin mediated cellular traction forces (high on the boundary) enabled trans-activation of Notch receptor.

Notch signaling is multifaceted because of the diverse interactions between the Notch receptors and the distinct Notch ligands. For this paper and the computational model, we focus on the Notch ligands Dll1 and Jag1 due to their contrasting functions in liver differentiation based on our previous work[13,15]. We have also previously shown upregulation of both these ligands during TGFβ-mediated biliary differentiation[13]. Further, it has been reported that the portal vein mesenchymal cells express Jag1, which mediates Notch signaling in the hepatoblasts lining the portal vein during biliary differentiation[2] and loss of Jag1 leads to bile duct paucity[42]. Hence, the suggestive experimental results in the context of liver differentiation represent a compelling system to explore the multifaceted nature of Notch signaling further using the computational model. Here, we report an advancement of a Notch modeling approach that incorporates enabling features such as the linking of two canonical notch signaling interactions, cis-inhibition and trans-activation, with biomechanics-mediated interactions with E-Cadherin, facilitated by the cellular microarray platform. The first phase of the computational model development was to

establish the simulated, 'in silico', knockouts for Dll1, Jag1, and E-Cadherin, and the assessment of the resulting Notch signaling patterns. Subsequently, the aggregate model was further optimized to support extrinsic perturbations, such as the treatment with exogenous growth factors. Notably, following both EGF and TGFβ2 treatments in the wild-type cells, there is an increase in biliary differentiation. However, with EGF treatment, the increased biliary differentiation was restricted to the outermost boundary region, whereas following TGFβ2 treatment, the biliary differentiation extended into the center of the cell domains. For E-Cadherin knockdown, slight restoration of the biliary pattern was observed following EGF treatment, and a significant increase in biliary differentiation across the whole island with the TGFβ2 treatment. The differentiation pattern alterations as a result of the growth factor treatments were also successfully emulated by the computational model to a large extent, with relatively simplified modifications to the model for each perturbation. Collectively, these results suggest a cooperative mechanism in which multiple cellular forces in combination with Notch signaling and morphogens can lead to the formation of distinct patterns of progenitor cell differentiation.

The development of the computational model was a cyclic process in which it had to be reiterated multiple times in order to corroborate it with the experimental data. Consequently, the systematic experimental perturbations were crucial for the

development of the integrated aggregate model. Each additional exogenous perturbation, such as exogenous growth factor treatment, led to a unique discovery about the multifaceted signaling. It also facilitated fine-tuning of various parameters that were used to simulate the exogenous knockdown or the growth factor treatment. There have been a number of other studies modeling various aspects of Notch signaling in the context of numerous biological systems[43–45] such as the formation of long-range patterns[46] and the complex Notch-Delta-versus-Jagged interactions[47]. Furthermore, there are multiple organoid-based models focused on a specific tissue system[48], and some studies have modeled biomechanical cues alongside signaling pathways[49]. However, such models have rarely been employed in the study of liver development and regeneration. Here, using a cell type that exhibits bipotential liver progenitor differentiation capacity, we tested multiple hypotheses of how different cellular forces might be affecting spatially dependent Notch signaling. In its current configuration, the aggregate computational model simulates the activity of Notch signaling (i.e. level of Notch Notch targets) as an extension of the NICD cleavage and as a collective simulation Notch target gene expression. However, it is likely that different downstream Notch target genes exhibit distinct expression magnitude shifts as a result of active Notch signaling. For instance, this may be one explanation for why Hes1 experimental fold-changes between cells in the interior versus the boundary are relatively smaller compared to drastic computational model changes and even the drastic OPN pattern observed. Overall, the quantification of intracellular OPN expression with immunostaining exhibited a much higher signal-to-noise ratio, and hence the computational model was correlated with the OPN pattern more closely. Future efforts aimed at advancing this model could be focused on the incorporation of specific downstream differentiation pathways and markers, such as OPN and HNF4α expression, integrated fully into the multitarget model. Plus, the incorporation of a broader range of mechanotransduction mediators as inputs could be further explored. For instance, the Hippo pathway and YAP/TAZ are implicated in both hepatocytes[50] and Notch signaling[51], which could represent a possible link in combining biomechanics with Notch signaling in hepatocytic differentiation.

In summary, we present a comprehensive experimental and computational model to study the cooperative effects of biomechanical cues and Notch signaling in the setting of liver differentiation using controlled microenvironments. The experimental system enabled both cell-intrinsic and exogenous perturbations of the differentiation patterning of liver progenitor cells within a controlled multicellular geometry, while the equivalent computational model served as the foundation for intriguing hypotheses of how underlying patterns of cellular mechanical stimuli intersect with Notch signaling activation. Future efforts can be aimed at the further investigation of the molecular mechanisms of receptor–ligand dynamics influenced by mechanical forces, the translation of the differentiation patterning analysis system to a 3D organoid model with human liver progenitor cells, as well as the incorporation of intersecting signaling pathways like the Hippo pathway, within the computational model. Additionally, towards the further development of increasingly in vivo-mimetic platforms, multiple cell-type systems that introduce non-parenchymal liver cell types, such as endothelial cells or fibroblasts, could provide physiologically relevant heterotypic cell interactions such as those present within the liver portal vein and ductal plate regions. Overall, such studies like pave the way for a better understanding of how complex patterns arise during development as a function of cellular forces and molecular mechanisms.

## Methods

**Preparation of polyacrylamide hydrogels**. Polyacrylamide (PA) hydrogels were prepared following previous protocols[52–54]. Briefly, 12 mm glass coverslips were etched by immersing them in 0.2 M NaOH (Sigma-Aldrich 415413-1L) for 1 h on an orbital shaker and then rinsing with dH2O. The coverslips were then air-dried and placed on a hot plate at 110 °C until dry. For silanization, the cleaned coverslips were immersed in 2% v/v 3-(trimethoxysilyl) propyl methacrylate (Sigma Aldrich 440159-500ML) in ethanol and placed on the shaker for 30 min, followed by a wash in ethanol for 5 min. The silanized coverslips were air-dried, and again placed on the hot plate at 110 °C until dry. For fabrication of hydrogels with specific elastic moduli, prepolymer solution in dH20 with 8% acrylamide (Sigma-Aldrich A3553-100G) and 0.55% bis-acrylamide (Sigma-Aldrich M7279-25G) was prepared to achieve elastic moduli of 25 kPa. The prepolymer solution was then mixed with Irgacure 2959 (BASF, Corp.) solution (20% w/v in methanol) at a final volumetric ratio of 9:1 (prepolymer:Irgacure). This working solution was then deposited onto Rainx (Amazon Rain-X 800002245) coated slides (20uL/ coverslip) and covered with silanized coverslips. The sandwiched working solution was transferred to a UV oven and exposed to 365 nm UV A for 10 min (240E3 µJ). The coverslips with the hydrogels attached to it were immersed in dH2O at room temperature for a day in order to remove excess reagents from the hydrogel substrates. Before microarray fabrication, hydrogel substrates were thoroughly dehydrated on a hot plate for ≥15 min at 50 °C.

**Microarray fabrication**. Microarrays were fabricated as described previously[14,55,56]. Collagen 1 for arraying was diluted in 2×ECM printing buffer. to a final concentration of 250 µg/mL and loaded in a 384-well V-bottom microplate. To prepare 2× ECM protein printing buffer, 164 mg of sodium acetate and 37.2 mg of ethylenediaminetetraacetic acid (EDTA) were added to 6 mL dH2O. After solubilization, 50 µL of pre-warmed Triton X-100 and 4 mL of glycerol were added. 40–80 µL of glacial acetic acid was added, titrating to adjust the pH to 4.8. A robotic benchtop microarrayer (OmniGrid Micro, Digilab) loaded with SMPC Stealth microarray pins (ArrayIt) was used to microprint ECM combinations from the 384 microwell plate to polyacrylamide hydrogel substrate, resulting in ~600 µm diameter arrayed domains. Fabricated arrays were stored at room temperature and 65% RH overnight and left to dry under ambient conditions in the dark.

**Cell culture and microarray seeding**. We utilized BMEL 9A1 cell line between passages 32 and 37. These cells were established in[57]. The cells were seeded on tissue culture plastic coated with collagen I (0.5 mg/ml) and subsequently cultured under controlled environmental conditions (37 °C and 5% CO2). Treatment with trypsin-EDTA (0.25% v/v) for 5 min was used to detach cells for subculturing. Basal growth media for expansion consisted of RPMI 1640 with fetal bovine serum (10% v/v, FBS), penicillin/streptomycin (1% v/v, P/S), L-glutamine (1% v/v), human recombinant insulin (10 µg/ml, Life Technologies, 12585–014), IGF-2 (30 ng/ml, PeproTech, 100–12), and EGF (50 ng/ml, PeproTech, AF-100–15). The Jag1 KD, and Dll1 KD cells were generated by lentiviral transduction with shRNA constructs targeting a Jag1, and Dll1, respectively, the details and validation of which we have described elsewhere[13]. Identical growth conditions were used for the knockdown cells as the wild-type cells described above for BMEL cell culture. The E-Cadherin knockdown and Negative Control (NC) transient cell line were created using lipid-based transfection of E-Cad siRNA (Thermofisher Scientific, AM16708, siRNA ID 161135) and nonsense siRNA (Thermofisher Scientific, AM4611). Lipofectamine RNAiMAX (Thermo Fisher Scientific 13778075) was used according to the protocol recommended by the manufacturer. Briefly, for 5 mL of media, 500 µL of transfection solution was made. 15 µL of the RNAiMax was dissolved in 250 µL of OptiMEM media and 14 µL of 10 µM stock of siRNA was dissolved in 250 µL of OptiMEM media. Both the solutions were mixed to get 500 µL of transfection solution and incubated at RT for 30 min. The transfection solution was added to 5 mL of growth media with cells at 50% confluency and was transfected for 24 h before harvesting for microarray differentiation studies. The primers used to confirm the knockdown using PCR are listed in Supplementary Table 4.

For the microarray differentiation studies, the polyacrylamide hydrogels were hydrated in 1X PBS 1% P/S solution and sterilized under the UV for 20 min. Cells were harvested at 70% confluency, and seeded at 100% confluency, 100e5 cells/ 12mm coverslip for microarray studies and 500e3 cells/dish for TFM. Differentiation media consisted of Advanced RPMI 1640 (Life Technologies, 12633–012) with FBS (2% v/v), P/S (0.5% v/v), L-glutamine (1% v/v), and minimum non-essential amino acids (1% v/v, Life Technologies, 11140–050). BMEL cells tested negative for several different species of Mycoplasma using PCR and gel electrophoresis. Cells were allowed to adhere to arrays for 6 h with intermittent manual gentle shaking every 20 min for the first 2 h. The arrays were washed twice with differentiation media and subsequent addition of experiment-specific treatments. The media was replenished every 48 h. EGF treatment was done at 50 ng/mL (0.025% of 0.2mg/mL stock of EGF in 0.2% BSA) and TGFb2 treatments was done at 0.1ng/mL (0.00025% of 20ug/mL stock TGFb2 in 4mM HCl). The growth factor treatments were started at 6 h post-seeding on the microarrays for 72 h until fixation.

**Immunostaining**. The BMELs were treated 2 h prior to fixing with Brefeldin A (10ug/mL,Tocris 1231), an inhibitor of protein translocation to Golgi, to retain OPN inside the cells. The cells were fixed with 4% w/v PFA in 1XPBS solution at 72-h timepoint starting after the 6 h seeding time. This was followed by a 10-min permeabilization step using 0.25%TritonX/1X PBS solution and a 45-min blocking step with the blocking buffer (5% donkey serum in 0.25%TritonX/1X PBS solution). Both the primary and secondary antibody staining was either done overnight at 4 °C or for 1 h at room temperature in dark, with 3x 5 min 1X PBS washes in the middle. The antibody cocktail was prepared in the blocking buffer. Primary antibody anti-HNF4a (ab41898) was used at 1:200 dilution, anti-OPN (AF808) was used at 1:50 dilution and anti-E-Cad (AF748-SP) was used at 1:50. Actin was stained using fluorescent antibody Acti-Stain (PHDH1-A) at 1:700 dilution. All secondary antibodies (anti-mouse: ab98795, anti-goat: ab96935) were used at 1:50 dilution. For 12 mm coverslips, 30 μL of antibody cocktail was used to stain. Samples were then mounted in DAPI Flourmount G (Southern Biotech 0100-20).

**ISH-HCR**. Protocol from ref. [24] was used. Briefly, custom probes were ordered from Molecular Instruments, Inc for Hes1 (Accession Number: NM_008235.2) designed for the B3 amplifier. The amplifier sequences B3 were also ordered from Molecular Instruments. Cells were fixed in 4%PFA in 1X PBS solution for 15 min. The cells were permeabilized in 0.1% Tween-20 1X PBS solution for 10 min at RT, followed by 10-min incubation with 5X SSC 0.1% Tween-20 (5X SSCT) buffer. The cells were incubated in the pre-hybridization buffer for 45 min at 37 degrees. The hybridization was by preparing a 10nM solution of the Hes1 probes in the hybridization buffer and incubating the cells overnight at 37 degrees. The cells were washed in 30% formamide wash buffer (30% formamide, 5X SSC, 9mM citric acid) 5 times for 5 min each. The cells were then washed in SSC/Tween/Heparin Wash Buffer (5X SSC, 0.1% Tween-20, 50 ug/mL Heparin) thrice for 5 min each. The samples were then incubated in amplification buffer for 45 min. Amplification solution with amplification buffer and B3 hairpin amplifier h1 and h2 were prepared at 30 nM concentration for each and incubated with the cells overnight. The cells were washed 5 times for 5 min each with 5X SSCT prior to mounting the cells with DAPI Flouromount G (Southern Biotech 0100-20). For analyzing the speckles in the boundary versus the edge, the edge was detected using Cell Profiler's[58] Find Neighbors function. The edge was identified as cells not surrounded by neighbors on all sides. The number of speckles were then identified (using Cell Profiler) and calculated in both the regions using R.

**Imaging and analysis**. The cellular microarrays for the immunostaining were imaged using Axioscan.Z1 Slide Scanner and ×10 objective. A wide tile region was defined for the whole array region which was then stitched offline using Zen and exported into TIFF Images for each individual channel. Images of entire arrays were converted to individual 8-bit TIFF files per channel (i.e., red, green, blue) by Fiji (ImageJ version 1.52p)[59]. The images were cropped in MATLAB (version R2018b) to separate each array in a single image. Positional information for each array was automatically calculated using their relative position from the positional dextran-rhodamine markers. CellProfiler (version 4.0.0)[58] was used get per cell measurement for each channel. Nuclei were identified using the DAPI channel image using IdentifyPrimaryObject module and other stain were associated with a specific nuclei was identified by looking at the red/green stain around these nuclei using IdentifySecondaryObject module. The MeasureObjectIntensity module was used to quantify single-cell intensity. The data were exported to CSV files that were then imported in RStudio for data visualization. For each island the centroid was determined using the coordinates of the nucleus found. For every cell in the island, the distance from the center was quantified and normalized by the top 5 percentile distance from the center to obtain a normalized radius value for each cell. All measurements are analyzed for each independent island, a minimum of four biological replicates (independent experiments) and a minimum of 15 technical replicates were used for every experimental measurement. For the line plot, 95% confidence interval is reported as a function of the radius on an island. For the boxplots, wilcox test was used to calculate significance of differences.

**Traction force microscopy**. A detailed protocol is mentioned with all the experimental and computational analysis details were taken from ref. [60]. Briefly, we fabricated the PA hydrogels in glass-bottom 35 mm Petri dishes (Cell E&G, GBD00002-200). This enabled us to perform TFM on live cells at 37 °C and 5% CO2. To measure the cell-generated forces, 1 μm far-red fluorescent beads (0.2% v/v, Life Technologies, F-8816) were added to the polyacrylamide solution and the hydrogels were made using the same protocol as described before. For seeding cells, 500e3 cells/dish were used and TFM was performed at 24-h timepoint. The arrays were live-imaged using (37 °C and 5% CO2) Axiovert 200M microscope (Carl Zeiss, Inc.). The microscope was used to capture phase contrast and far-red fluorescent images to record cellular position and morphology along with bead displacement before and after cell dissociation with sodium dodecyl sulfate (1% v/v in 1× PBS). The captured images were analyzed to calculate traction forces in MATLAB software (MathWorks, Inc) code available at[60].

**Western Blot**. Cell pellets from 6-well plates were collected and resuspended in sample buffer (1x NuPage NP0007 (ThermoFisher Scientific), 2.5% β-

mercaptoethanol). Samples were heated for 5 min at 95 °C, sonicated, and heated at 95 °C for another 5 min. Lysates were then loaded in NuPAGE™ 4–12%, Bis-Tris, 1.0–1.5 mm, Mini Protein Gels (ThermoFisher Scientific, NP0321BOX) and run for 1 h at 130 volts. The proteins were transferred to a nitrocellulose membrane (BioRad, 1620215) for 1.5 h at 70 V using Mini PROTEAN 3 cell (BioRad, 525BR). The membranes were blocked for 1 h in 5% dried milk in TBS-T and then washed 3 times with TBS-T before incubation with primary antibodies. Primary antibodies for Jagged1 (Cell Signaling, 3195T), E-cadherin (Cell Signaling, 2620T), and GAPDH (Cell Signaling, 2118S) were diluted 1/200, 1/200, and 1/1000, respectively, in TBS-T with 1% dried milk and incubated with the membrane overnight at 4 °C. Next, the membranes were washed 3 times with TBS-T prior to incubation with IRDye® 680RD Donkey anti-Rabbit IgG Secondary Antibody (Licor, 926-68073) for 1 h at room temperature. Samples were washed again 3 times with TBS-T and imaged with a LI-COR odyssey FC imager. Fluorescence intensity was quantified using ImageJ.

**Computational model**. To generate the circular cellular island domains for simulation, we first created a 400 unit polygon mesh of a roughly circular area using PolyMesher, a publicly available algorithm written for MATLAB[61]. From this mesh, we used customized MATLAB scrips to create an input file for the Surface Evolver software package[62]. Surface Evolver is a program that minimizes an assigned energy function for a 2D or 3D surface conformation, given a set of initial conditions and constraints. Here we assigned a surface tension to the elements in the polygonal mesh and used the included evolve and refinement functions to minimize energy subject to the constraints of the circular island boundary and equal area for all mesh units. Additionally, the surface tension for elements at the outer boundary was set as three times that of the others. This "evolved" the polygon mesh to the lowest energy form, a circular domain made of discrete unit "cells" of equal area, which approximates the cellular island geometry. We then exported the element list, node coordinate list, and connectivity matrix, which were used as inputs to the simulation.

The MATLAB code for Notch signaling was obtained from[25] and modified. The modified equations for the base model are specified in Eqs. (1)–(5). Experimental traction values for each of the three gene knockdowns were measured, normalized and incorporated in the computational model. For each biological experiment, the average maximum stress was obtained among all the biological condition and the traction was normalized to this average maximum stress. All the biological experiments were averaged and incorporated as is in the equation in the models. The equations were simulated cells in a circular lattice using ode15 function on MATLAB. For the perturbation simulation the parameters were simulated as listed in Supplementary Table 2.

**Statistics and reproducibility**. All microarray experiments consisted of at least three biological replicates, with 15 technical replicates, or islands, per biological replicate per combination of gene knockdown, treatment, and readout. For comparison between conditions in this study, Wilcoxon tests were performed using the wilcox.test function in R. $P$ values of <0.05 were considered significant. For the line graph demonstrating quantification of a readout as a function of the radius, 95% confidence interval was calculated and displayed using the geom_smooth function in R.

**Reporting summary**. Further information on research design is available in the Nature Research Reporting Summary linked to this article.

## Data availability

The raw data is supplied in Supplementary Data 1. The unedited western blot images are provided in Supplementary Fig. 9.

## Code availability

All code files with instructions and simulation results are in the following box folder: Jain_et_al_MATLABCode | Powered by Box

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

## Acknowledgements

This work was supported by the National Institutes of Health (R01DK125471 to GHU). The authors acknowledge Helene Strick-Marchand and Mary C Weiss (Institut Pasteur)

for providing BMEL cells. The authors also thank the Institute of Genomic Biology's core facility at the University of Illinois at Urbana-Champaign for their assistance with the microscopy performed in these studies.

## Author contributions

Conceptualization, I.J. and G.H.U.; Methodology, I.J., P.P.P., and G.H.U.; Investigation, I.J., I.C.B., A.A., M.B., and N.G.; Resources, I.J., I.C.B., and G.H.U.; Writing and Editing, I.J., P.P.P., and G.H.U.; Funding Acquisition and Supervision, G.H.U.

## Competing interests

The authors declare no competing interests.
