## [Peer Review File · Communications Biology]

Reviewers' comments:

Reviewer #1 (Remarks to the Author):

In the manuscript "Delineating Effects of Notch and Biomechanical Signals on Patterned Liver Differentiation using Modeling and Cellular Microarrays" Jain and colleagues investigate the relationship between biophysical forces and Notch signalling in the context of liver cells differentiation. The authors propose a hybrid approach which leverages both on the experimental characterisation of the system, through ISH-HRC and high throughput microarrays, and on its mathematical modelling through a system of Ordinary Differential Equations. The model represents a significant extension of similar approaches available in the literature and has been constructed to recapitulate the outcomes of perturbation experiments designed to reveal the interactions between the constituents of the system. I think the manuscript is well written, and I appreciate the interdisciplinary paradigm followed by the authors to investigate this relevant aspect of cell differentiation. Few comments that I believe are important to further improve the quality of the study follow, together with some points relative to secondary aspects of minor relevance.

Majors:

1) The possibility to reproduce the results presented in an article is essential both for its revision, and for the community to get the most from the work of the authors. For this reason, I encourage the authors to provide access to the data generated for their study both experimentally and in-silico. Indeed, the folder "Jain_et_al_Data" is empty, and the folder "Jain_et_al_MATLABCode" does not contain the results of the simulations. A final minor point, I really appreciate that the MATLAB script has been well commented, I would just add a readMe.txt file to clearly specify the content of the directory.

2) Perturbation experiments based on the knockdown (KD) of DLL1, JAG1 and E-CAD are essential in the economy of this work, however, the depletion of these genes have been tested only at the RNA level and in single replicate. First, the PCR experiments should be repeated, and a statistical comparison against the control should be performed. Second, the authors should check the KD efficiency at the protein level (e.g. through western blots). These experiments will provide a more solid quantification of the effect of siRNAs/shRNAs, which can be included in the simulation step instead of the arbitrary values currently used (0.70 and 0.75).

3) The model proposed in this work depends on a remarkable number of parameters which are presented through the text and in Supplemental Table 1. I think it would be important to extend the table including, for each parameter, the rationale behind its selection with relevant references to the literature when required.

Minors:

Lines 111-113 and 118-119: The same condition ($0.95 \leq \text{Radius} \leq 1$) is reported for both boundary and interior cells.

Line 131: The authors mention the correlation between traction force and biliary differentiation (OPN+ cells) but this can be numerically estimated. More in general, all the quantities profiled over the radial distance from the centre could be compared with a correlation measure to provide a general overview of their similarity which can complement the interior/boundary comparison.

Lines 165-166: The introduction of parameters of the model before its formulation is confusing.

Line 275: I would substitute/complement terms as "much higher" with quantitative comparisons.

Line 377: I would extend the description of how the authors optimised ISH-HCR for the microarray platform.

Figure 1: (B) I would report a vertical line to show the 0.95 threshold (same for similar panels in the other figures). Moreover, the panel is surrounded by a grey line which should be removed

(same for many other figures). (D) The figure is low-quality (same for 4D). (E) The label "25" is cut (same for other figures like 6B, 6G and 6M), and the caption reports a unit for the traction force different than the one shown in the figure; kPa and Pa respectively.

Figure 3: (E) the scale is not readable, the font size should be increased (same for 4G, 5H and 7D). Moreover, the label for the radius should be uniform across panels (with or without the "dimensionless" specification).

Figure 6: The figure is too small and difficult to read if printed in a standard A4 format.

Reviewer #2 (Remarks to the Author):

The authors present a study where both in vitro experimental and computational modeling methods were used to study heterogeneous hepatocyte cell differentiation patterning in bipotential mouse embryonic liver (BMEL) cells based on key factors of the intercellular Notch signaling pathway and biomechanical force effects. By growing BMEL cell monolayers within circular hydrogel wells to approximate the geometry of the ductal plate tissue surrounding the portal vein, the authors were able to study BMEL differentiation patterns under a number of knockdown or stimulation conditions. This study builds on previous studies of cellular differentiation in circular geometries by including an ordinary differential equation-based computational model to gain new insights into the mechanisms of interest. Applying the model to a 2D hexagonal lattice, authors were able to recreate experimentally-observed results. The experimental results are interesting and convincing, and support the assertions of Notch and the studied downstream molecules playing a role in the heterogeneous differentiation of BMEL cells. The model is interesting, but key information is missing, including initial and boundary conditions, and clear descriptions of how the model was perturbed to replicate each experimental condition and estimates of model stability to these perturbations, reducing enthusiasm for these results. The paper is logically well-presented, but numerous grammatical errors make it challenging to follow in some places, and parts appear to have been prepared in a rush. The referee's assessment is the paper shows promising results, and believes it will be of interest to the scientific community, but some revisions can further strengthen its impact.

Some revisions are necessary before publication, these are detailed below.

Major comments:

1. The authors assessed cell traction forces via live TMF based on live imaging of deformation in a 1 μ m far-red fluorescent bead labeled hydrogel, and report that traction forces were highest around the physical edge of the island in the microarray. This is likely, in part, an artifact of the lack of cells outside the boundary that would serve to balance the cell-cell adhesion and contractile forces acting in the direction towards $R=0$ in the cell island. This will result in an artificially large displacement between the fluorescent beads directly inside the outer radius of the cell island vs. those directly outside at $R > 1.0$ (see also comment #2). This larger displacement is likely associated with a similar overestimation of adhesion force at this location. Do the authors think the modeled forces at this boundary are reasonable based on this approximation? At minimum, some discussion of this would be interesting.

Although the referee has found drawing the force balance impractical in text format, they hope the following may help clarify further. Cells might be assumed to impose similar contractile forces on the substrate, and we can expect a roughly linear contraction across the radius increasing from the zero displacement assumption at $R=0$ (that is, uniform force balance in all radial directions); this is clearly seen in Figure 1E, $R < \sim 0.7$. However, as we get closer to the edges, the force imbalance becomes skewed with a greater magnitude towards $R=0$ due to fewer cells to resist these forces, as shown in the increasing slope in Fig 1E $R > \sim 0.7$. At the $R=1$ boundary, the magnitude of this force imbalance is maximized. This will cause the largest displacement in the substrate at $R=1$; and displacement between fluorescent beads should be expected to be largest at this location towards the radius; thus the substrate is "stretched" more at this location (due to lack of cell contractile forces on the hydrogel at $R > 1$).

2. The force balance in the radial substrate displacement problem across the circular (symmetrical) island implies a displacement=0 boundary condition at $R=0$. Thus, the force balance at this point should sum to zero in all directions (otherwise there would be displacement in the direction of the force). Although cell heterogeneity may cause slight deviations when measured, averaging over many tests should come close to the known zero-displacement condition (especially as 1 μ m labeled beads (~10% cell diameter) should give roughly cell-scale resolution in TFM). Figure 1D,E shows that the force measured at $R=0$ is almost 1/3 that at the boundary, showing that this was not the case. Can the authors comment on this? A brief addition to the discussion would be interesting.

3. Eq (1), Eq (3): should activation and inhibition terms have opposite signs? Rate constants k_{td} and k_{cd} are shown both positive in SI. If a correction is needed, then please also review Eqns 8 and 9.

4. Initial and boundary conditions for the model equations are not provided.

5. The authors present the idea of cellular islands in the model, which is a key feature of the work presented. This should be clearly defined in the first place the concept of "islands" appears (line 136)

6. Lines 247-250: traction force gradient is likely a combination of 1) contractile forces from each individual cell on their local substrate and 2) from the cell-cell adhesion forces. Do you think the local contractile forces from each individual cell could explain part of this observed outcome?

7. Discussion about the lack of cells surrounding the island boundary vs. a physical system (e.g., an array of ductal plates) would be interesting. What does this imply about molecular boundary conditions and local forces in a full liver tissue? For example, each ductal plate might be in compression in vivo, with higher compression (hoop-stress like) forces on outer layers. Does this imply these results would translate well to the in vivo system? Could a future paper involve patterning the island close together to approximate a larger section of liver tissue?

8. It would be helpful to have a table showing model parameters that were perturbed for the various model results shown in the figures, along with values or value ranges they were perturbed to across the "n>15 technical replicates" reported

9. As a follow-up to #8, how sensitive was the model to these perturbations? This information would help assess the reliability of the results shown. Are these robust to perturbation within a reasonable range, or are the results shown the best match from a very specific scenario that may not be generally applicable? This would help answer these questions. There are several standard sensitivity analysis methods to address this problem

10. Definitions of model variables can be presented in a more clear way so the model equations may be more easily understood. For example, lines 177-178 appear to list 10 variables as written; this could be clarified as "Notch receptor (NRi), Dll1 (Di), ...". Some variables (e.g., Cneighbor, Si) are defined so much farther down the referee initially assumed that the authors forgot to define them, they could be moved closer to where they first appear.

11. Some naming conventions seem to be inconsistent, for example "HNF4alpha" in the main text vs "HNF4a" in e.g., Figs 1,3-5; "Dll1" vs. "DLL1"; "Jag1" vs. "JAG1", h vs. hours, etc.. Authors should review carefully for others as well.

Minor comments:

1. Line 113: please correct "in the interior ($0.95 \leq \text{Radius} \leq 1$)". Should this be " $\text{Radius} \leq 0.95$ "?
2. Same for boundary and interior radius, lines 118-119
3. Line 184: it is unclear how "various parameter are further explained" in SI figure 2
4. Line 214: may want to define the acronym KD here: "...by knocking down (KD)..."
5. Line 217: $0.95 \leq \text{Radius} \leq 1$?

6. Line 475: maybe -> may be
7. Figure 1: it could be helpful to show a gridline at $x=0.95$ on appropriate plots (as appropriate in other figures too) to help show the defined interior:exterior boundary. This comment is very minor but is included here in case it's helpful
8. Fig 3, 4 captions: there's no Blue: DAPI shown in these figures
9. Figure 3F, 5I: WT simulated targets appears to drop <0 around $R=0.75$; please review. This appears to also be the case for Fig 4H: NC curve. If this is a result of numerical approximation it should be explained
10. Authors should carefully review all figures to make sure panels haven't overlapped and placement is as intended. For example, the legend of Fig 6G and "1000" on Fig 6M y-axis.
11. Line 664: "0.2 N NaOH" -> 0.2 M NaOH?
12. Line 757: biological

Reviewer #3 (Remarks to the Author):

The authors have tried to identify the role of Notch signalling and biomechanical cues during liver differentiation using both microarray and modelling approaches.

The claims made by the authors are well supported by both the experimental and model outcomes.

I recommend this work for publication without further corrections.

We thank the reviewers for their thorough review and well organized, helpful comments. Please find our point-by-point responses below which also indicate the relevant revisions made to the manuscript in response. In the revised manuscript provided, changes are **highlighted in blue font**. Small changes which correct grammatical errors, or improve readability, have also been made throughout the manuscript and may not be highlighted in blue. Changes made to content and to directly address reviewer comments have been highlighted.

Reviewer #1 (Remarks to the Author): In the manuscript "Delineating Effects of Notch and Biomechanical Signals on Patterned Liver Differentiation using Modeling and Cellular Microarrays" Jain and colleagues investigate the relationship between biophysical forces and Notch signalling in the context of liver cells differentiation. The authors propose a hybrid approach which leverages both on the experimental characterisation of the system, through ISH-HRC and high throughput microarrays, and on its mathematical modelling through a system of Ordinary Differential Equations. The model represents a significant extension of similar approaches available in the literature and has been constructed to recapitulate the outcomes of perturbation experiments designed to reveal the interactions between the constituents of the system. I think the manuscript is well written, and I appreciate the interdisciplinary paradigm followed by the authors to investigate this relevant aspect of cell differentiation. Few comments that I believe are important to further improve the quality of the study follow, together with some points relative to secondary aspects of minor relevance.

Majors:

1) The possibility to reproduce the results presented in an article is essential both for its revision, and for the community to get the most from the work of the authors. For this reason, I encourage the authors to provide access to the data generated for their study both experimentally and in-silico. Indeed, the folder "Jain_et_al_Data" is empty, and the folder "Jain_et_al_MATLABCode" does not contain the results of the simulations. A final minor point, I really appreciate that the MATLAB script has been well commented, I would just add a readMe.txt file to clearly specify the content of the directory.

Response: The raw data for the experimental results have been added to the Jain et al Data folder linked from the manuscript. For the computational model, the simulation results and a readMe text have also been added to the MATLAB code folder.

2) Perturbation experiments based on the knockdown (KD) of DLL1, JAG1 and E-CAD are essential in the economy of this work, however, the depletion of these genes have been tested only at the RNA level and in single replicate. First, the PCR experiments should be repeated, and a statistical comparison against the control should be performed. Second, the authors should check the KD efficiency at the protein level (e.g. through western blots). These experiments will provide a more solid quantification of the effect of siRNAs/shRNAs, which can be included in the simulation step instead of the arbitrary values currently used (0.70 and 0.75).

Response: We acknowledge that quantification of the various knockdown cells was previously performed using only a single replicate PCR experiment. To address this, we have now included 3 biological replicates for the PCR for every knockdown, with statistical comparison to the control cells using a t-test. We have also performed Western Blot for E-Cadherin and Jagged1 knockdown in BMELs, quantification of which is included in Supplementary figure 4. Due to a combination of low expression and unavailability of appropriately sensitive Western blotting-compatible antibodies, expression of Dll1 protein, cannot be

effectively evaluated using Western Blot. For E-Cadherin knockdown, we observe approximately 95% knockdown of both the mRNA and of the resulting protein. For Jagged 1 knockdown in BMELs, a 50% knockdown is achieved in the mRNA expression which leads to about 90% knockdown in the protein expressed. For Dll1 knockdown in the BMELs, a 75% knockdown is achieved in the mRNA expression. The simulation values for the knockdown of these pathway proteins are representative of both the reduced protein expression level and function of the respective factor. These simulation values were determined based on how each knockdown was resulting in the simulated notch target pattern, and if that pattern matched to the experimentally observed OPN pattern. 70% for E-Cadherin knockdown and 75% knockdown for Jag1 and Dll1 is appropriate as it most effectively emulated the resulting pattern for the base model and for the growth factor treatments.

3) The model proposed in this work depends on a remarkable number of parameters which are presented through the text and in Supplemental Table 1. I think it would be important to extend the table including, for each parameter, the rationale behind its selection with relevant references to the literature when required.

Response: That is a very important point, thank you for pointing it out. The current model has been adapted from reference 25. All the original parameter in this paper were used initially for the base model. As we added perturbations to the systems, some of the parameters in the base model had to be diligently modulated to match all perturbations to the experimental result obtained. To explain the process better, supplemental figure 2 has 4 different values for two parameters β_{N_R} and β_{NICD} , which when tuned, changed the pattern of the simulated active notch targets. Further, a supplementary figure (Supp. Fig. 8) has been added where we demonstrate how the simulated active notch targets change with changing the parameter values, specifically for those parameters that were used for the various in silico knockdown and growth factor treatment perturbations. We have also added a brief explanation of parameter value selection in the main text in lines 218-226

Minors:

Lines 111-113 and 118-119: The same condition ($0.95 \leq \text{Radius} \leq 1$) is reported for both boundary and interior cells.

Response: Thank you for bringing this to our attention. This typo has been fixed in the main text.

Line 131: The authors mention the correlation between traction force and biliary differentiation (OPN+ cells) but this can be numerically estimated. More in general, all the quantities profiled over the radial distance from the center could be compared with a correlation measure to provide a general overview of their similarity which can complement the interior/boundary comparison.

Response: This is a great point. We have included correlation plots with metrics in Supplementary figure 1. A correlation value of 0.8 is observed for radial OPN intensity and Normalized Traction Force. Similarly, a correlation factor of 0.8 was observed for radial OPN intensity and radial Ecad Intensity.

Lines 165-166: The introduction of parameters of the model before its formulation is confusing.

Response: We appreciate the feedback. We have re-written the explanation of the model is highlighted in the main text lines 183-198.

Line 275: I would substitute/complement terms as "much higher" with quantitative comparisons.

Response: Thank you for pointing this out. We have replaced the terms with estimation of the difference.

Line 377: I would extend the description of how the authors optimised ISH-HCR for the microarray platform.

Response: The original motive for adding this in the discussion was to highlight the extended application of the microarrays for looking at spatial transcription. The existing protocol by <https://pubmed.ncbi.nlm.nih.gov/29945988/> was implemented. We have now moved the details to the methods section in lines 803-806.

Figure 1: (B) I would report a vertical line to show the 0.95 threshold (same for similar panels in the other figures). Moreover, the panel is surrounded by a grey line which should be removed (same for many other figures). (D) The figure is low-quality (same for 4D). (E) The label "25" is cut (same for other figures like 6B, 6G and 6M), and the caption reports a unit for the traction force different that the one shown in the figure; kPa and Pa respectively.

Response: Thank you for pointing these details. We have addressed them and fixed the figures.

Figure 3: (E) the scale is not readable, the font size should be increased (same for 4G, 5H and 7D). Moreover, the label for the radius should be uniform across panels (with or without the "dimensionless" specification).

Response: Thank you for pointing this out, it has been fixed in all the figures.

Figure 6: The figure is too small and difficult to read if printed in a standard A4 format.

Response: We appreciate the feedback and the figure has now been divided in two figures for better readability.

Reviewer #2 (Remarks to the Author):

The authors present a study where both in vitro experimental and computational modeling methods were used to study heterogeneous hepatocyte cell differentiation patterning in bipotential mouse embryonic liver (BMEL) cells based on key factors of the intercellular Notch signaling pathway and biomechanical force effects. By growing BMEL cell monolayers within circular hydrogel wells to approximate the geometry of the ductal plate tissue surrounding the portal vein, the authors were able to study BMEL differentiation patterns under a number of knockdown or stimulation conditions. This study builds on previous studies of cellular differentiation in circular geometries by including an ordinary differential equation-based computational model to gain new insights into the mechanisms of interest. Applying the model to a 2D hexagonal lattice, authors were able to recreate experimentally-observed results. The experimental results are interesting and convincing, and support the assertions of

Notch and the studied downstream molecules playing a role in the heterogeneous differentiation of BMEL cells. The model is interesting, but key information is missing, including initial and boundary conditions, and clear descriptions of how the model was perturbed to replicate each experimental condition and estimates of model stability to these perturbations, reducing enthusiasm for these results. The paper is logically well-presented, but numerous grammatical errors make it challenging to follow in some places, and parts appear to have been prepared in a rush. The referee's assessment is the paper shows promising

results, and believes it will be of interest to the scientific community, but some revisions can further strengthen its impact.

Some revisions are necessary before publication, these are detailed below.

Major comments:

1. The authors assessed cell traction forces via live TMF based on live imaging of deformation in a 1 μ m far-red fluorescent bead labeled hydrogel, and report that traction forces were highest around the physical edge of the island in the microarray. This is likely, in part, an artifact of the lack of cells outside the boundary that would serve to balance the cell-cell adhesion and contractile forces acting in the direction towards $R=0$ in the cell island. This will result in an artificially large displacement between the fluorescent beads directly inside the outer radius of the cell island vs. those directly outside at $R > 1.0$ (see also comment #2). This larger displacement is likely associated with a similar overestimation of adhesion force at this location. Do the authors think the modeled forces at this boundary are reasonable based on this approximation? At minimum, some discussion of this would be interesting.

Although the referee has found drawing the force balance impractical in text format, they hope the following may help clarify further. Cells might be assumed to impose similar contractile forces on the substrate, and we can expect a roughly linear contraction across the radius increasing from the zero-displacement assumption at $R=0$ (that is, uniform force balance in all radial directions); this is clearly seen in Figure 1E, $R < \sim 0.7$. However, as we get closer to the edges, the force imbalance becomes skewed with a greater magnitude towards $R=0$ due to fewer cells to resist these forces, as shown in the increasing slope in Fig 1E $R > \sim 0.7$. At the $R=1$ boundary, the magnitude of this force imbalance is maximized. This will cause the largest displacement in the substrate at $R=1$; and displacement between fluorescent beads should be expected to be largest at this location towards the radius; thus the substrate is “stretched” more at this location (due to lack of cell contractile forces on the hydrogel at $R>1$).

Response: The reviewer is correct that the traction force pattern observed is a consequence of the constraining geometry of the circular island. In the confluent cell island, all cells are connected to the surface and to each other and contracting in a similar fashion. As the reviewer notes, when the cells contract, the reaction forces are provided by the neighboring cells and the substrate. Towards the center of the island (near $R = 0$) the contractile forces are balanced against neighboring cells via cell-cell contacts. At the boundary, there are no neighboring cells, thus the contractile forces must be balanced against the substrate as cell to substrate traction force. The large displacements at the boundary are not an artifact of the boundary, but a true consequence of the fact that contractile forces at boundary are resolved as cell- to-substrate traction force, instead of cell-to-cell tension, which is a consequence of the boundary condition. It is postulated that the cells towards the center are not necessarily generating less contractile force, but that they are resolving these contractile forces as cell-to-cell tension, whereas towards the boundary, cells resolve more force as cell to substrate traction stress. Others in the literature have noted that this behavior is well modeled by approximating the cell sheet as a uniformly contracting elastic material which replicates this concentration of both displacement (substrate stretch) and cell to substrate traction stress. This has been shown demonstrated analytically (<https://journals.aps.org/prl/abstract/10.1103/PhysRevLett.107.128101#fulltext>), and using finite element models (<https://elifesciences.org/articles/38536> , <https://www.pnas.org/doi/10.1073/pnas.0502575102>). These references have been added in the main text along with our findings in lines 129-130, 211-215.

2. The force balance in the radial substrate displacement problem across the circular (symmetrical) island implies a displacement=0 boundary condition at $R=0$. Thus, the force balance at this point should sum to zero in all directions (otherwise there would be displacement in the direction of the force). Although cell heterogeneity may cause slight deviations when measured, averaging over many tests should come close to the known zero-displacement condition (especially as 1 μ m labeled beads (~10% cell diameter) should give roughly cell-scale resolution in TFM). Figure 1D,E shows that the force measured at $R=0$ is almost 1/3 that at the boundary, showing that this was not the case. Can the authors comment on this? A brief addition to the discussion would be interesting.

Response: The reviewer is correct to note that in the cell island, the force balance should sum to zero, as there is no cellular motion out of the circular boundary. As noted, here we are averaging traction force profiles across several islands, as well as averaging the behavior within each island, which will cause variation in the measurement. Further, in these plots we are describing the magnitude of the cell to substrate traction stress as opposed to a vector representation and relating these cell to substrate traction force magnitudes with notch and other signaling. Thus, summing and averaging of magnitudes would not average out stresses in opposing directions, thus not resulting in a 0 force sum. In general, we expect that the cell contractions result in stresses purely in the inward (towards the center direction). Indeed, in the idealized case, traction stresses are exclusively towards the interior and very tightly localized to the periphery. However, this arises only when 1) cells are perfectly connected to each other and thus acting as single sheet, and when 2) all cells are contracting uniformly. As shown in (<https://journals.aps.org/prl/abstract/10.1103/PhysRevLett.107.128101#fulltext>), if a region of cells is more or less contractile, this can result in traction stress in opposing directions. Such changes in contractility could arise during the differentiation occurring. Others in literature have shown that when cell-cell adhesions are disrupted, individual cells resolve contractile forces at their own periphery rather than to neighboring cells (<https://www.pnas.org/doi/10.1073/pnas.1217279110>). In our measurement, these randomly directed forces would add non-zero traction stress across the interior. Some cell-cell disruptions may be arising in our control cases. We note that in the case of Ecad knockdown, there are higher relative traction stress magnitudes across the interior which is consistent with this working model.

3. Eq (1), Eq (3): should activation and inhibition terms have opposite signs? Rate constants k_{td} and k_{cd} are shown both positive in SI. If a correction is needed, then please also review Eqns 8 and 9.

Response: The reviewer is correct in pointing out that that k_{td} and k_{cd} are corresponding to different types of Notch interactions. However, in the model the value of these parameters signifies the just absolute strength of interaction of Notch receptor and ligand. The effect it has in inhibiting/activating cleavage of NICD is modeled in Equation 2 where only the trans-activation term (with k_{td}) is enabling the cleavage. In all equation 1, 3, 4 the terms with these parameters signify the decrease in Notch Receptor and ligand concentration after they bind to each other since this receptor ligand complex is shortly endocytosed and digested as modeled by https://link.springer.com/protocol/10.1007/978-1-4939-1139-4_22. A brief explanation has been added in lines 188-192.

4. Initial and boundary conditions for the model equations are not provided.

Response: A table for the initial conditions has been added as the Supplementary Table 3. The initial condition for each cell had low values of both Jag1 and Dll1 with some noise, betaN levels of notch receptor and 0 levels of NICD and notch target genes, adapted from https://link.springer.com/protocol/10.1007/978-1-4939-1139-4_22. The exact functions of these initial

conditions could also be found in the getIC function in MATLAB code. This has also been explained in the main text from lines 226-230.

5. The authors present the idea of cellular islands in the model, which is a key feature of the work presented. This should be clearly defined in the first place the concept of “islands” appears (line 136)

Response: Thank you for pointing this out, a brief explanation of the islands has now been included in the beginning of the results, lines 112-114.

6. Lines 247-250: traction force gradient is likely a combination of 1) contractile forces from each individual cell on their local substrate and 2) from the cell-cell adhesion forces. Do you think the local contractile forces from each individual cell could explain part of this observed outcome?

Response: Yes, the traction force gradient is a function of the cell-cell adhesion and local contractile forces. With the E-Cadherin knockdown, since there was a decrease in the bipotential differentiation, where the OPN pattern was lost, we initially hypothesized that it was due to the loss of the traction force gradient. The knockdown of E-Cadherin could have led to reduced cell-cell adhesion, in which case there would be constant traction force across the whole island. However, we observed that the traction force gradient was not significantly affected, hinting that the cell-cell adhesion was not affected. This could have been due to the presence of other cadherin proteins, and this aspect is a topic of future examination. Notably, this initially counterintuitive observation led to our reexamination of potential E-cadherin-mediated mechanisms and refinement of the computational model. The explanation of these aspects in the main text has been revised for better understanding, in lines 280-288.

7. Discussion about the lack of cells surrounding the island boundary vs. a physical system (e.g., an array of ductal plates) would be interesting. What does this imply about molecular boundary conditions and local forces in a full liver tissue? For example, each ductal plate might be in compression *in vivo*, with higher compression (hoop-stress like) forces on outer layers. Does this imply these results would translate well to the *in vivo* system? Could a future paper involve patterning the island close together to approximate a larger section of liver tissue?

Response: The relevance of the microarray system with the *in vivo* development of bile ducts is a very valid question. The current microarray system is a rather simplified system to study how mechanical force gradients might be affecting cellular differentiation via complicated signaling pathways such as the Notch signaling. In the future, we do plan to incorporate other cell types around the circular patterns to emulate the *in vivo* contractile forces and cell heterogeneity. We also plan to further advance a 3D platform with defined shapes where we can delineate effect of mechanical in 3D. A brief discussion has been added on this topic in lines 531-534.

8. It would be helpful to have a table showing model parameters that were perturbed for the various model results shown in the figures, along with values or value ranges they were perturbed to across the “n>15 technical replicates” reported.

Response: The technical replicates stated in each figure legend is in reference to the biological experiments. The computational model was simulated in 400 cells in arranged in a circular lattice. The parameters for the base model are given in Supplemental Table 1. The initial concentrations for each cell of ligand, receptors and Notch targets are given in Supplemental Table 3. For every perturbation

parameter was changed to one other value shown in Supplementary Table 2. A brief explanation of this has been added to lines 218-230.

9. As a follow-up to #8, how sensitive was the model to these perturbations? This information would help assess the reliability of the results shown. Are these robust to perturbation within a reasonable range, or are the results shown the best match from a very specific scenario that may not be generally applicable? This would help answer these questions. There are several standard sensitivity analysis methods to address this problem.

Response: Thank you for pointing out this crucial part of the integrity of the computational model. We have included how simulated notch target value changes in the boundary cells (50 cells in the computational model) varies with parameter values of betaD, betaJ, ECad parameter (ECadp), egf and tgf across well-defined ranges (Supplemental Figure 8).

10. Definitions of model variables can be presented in a more clear way so the model equations may be more easily understood. For example, lines 177-178 appear to list 10 variables as written; this could be clarified as “Notch receptor (NRi), Dll1 (Di), ...”. Some variables (e.g., Cneighbor, Si) are defined so much farther down the referee initially assumed that the authors forgot to define them, they could be moved closer to where they first appear.

Response: Thank you for this feedback. The suggested change has been made and the section has been rewritten for better readability.

11. Some naming conventions seem to be inconsistent, for example “HNF4alpha” in the main text vs “HNF4a” in e.g., Figs 1,3-5; “Dll1” vs. “DLL1”; “Jag1” vs. “JAG1”, h vs. hours, etc.. Authors should review carefully for others as well.

Response: Thank you for this point. We have made the abbreviations consistent to the best of our knowledge.

Minor comments:

1. Line 113: please correct “in the interior ($0.95 \leq \text{Radius} \leq 1$)”. Should this be “ $\text{Radius} \leq 0.95$ ”?

2. Same for boundary and interior radius, lines 118-119

Response: These typos have been corrected

3. Line 184: it is unclear how “various parameter are further explained” in SI figure 2

Response: Additional explanation has been added in the main text, lines 218-226

4. Line 214: may want to define the acronym KD here: “...by knocking down (KD)...”

Response: This change has been incorporated in the main text, thank you for bringing it to our attention.

5. Line 217: $0.95 \leq \text{Radius} \leq 1$?

Response: This change has been incorporated in the main text, thank you for bringing it to our attention.

6. Line 475: maybe -> may be

Response: This change has been incorporated in the main text, thank you for bringing it to our attention.

7. Figure 1: it could be helpful to show a gridline at $x=0.95$ on appropriate plots (as appropriate in other figures too) to help show the defined interior:exterior boundary. This comment is very minor but is included here in case it's helpful

Response: We have now incorporated a vertical line at $x=0.95$ in all the OPN and HNF4a line graph measurement across all the figures, to augment the observations of the box blots of interior/boundary. Thank you for this suggestion.

8. Fig 3, 4 captions: there's no Blue: DAPI shown in these figures

Response: This change has been incorporated in the legend of the figure, thank you for bringing it to our attention.

9. Figure 3F, 5I: WT simulated targets appears to drop <0 around $R=0.75$; please review. This appears to also be the case for Fig 4H: NC curve. If this is a result of numerical approximation it should be explained

Response: The equations are solved using a MATLAB function ode15, which solves the equation numerically, and hence sometime near zero, we get some negative values for the NICD levels, which is oscillation around mostly around 0. Below is the raw data for simulated Notch Targets (NT) on y-axis in NC condition in 5I, which is zoomed in near 0 values of simulated notch targets. The negative values are very close 0, the lowest it goes is $-1e-11$. It is however interesting that this oscillation/ instability happens at $r=0.9$. We speculated that since there are opposing force trends on the circular islands, the traction force and cell-cell forces, with them being modelled as the converse of each other in the computational model. The complimentary force functions intersect at 0.9, resulting in instability around low levels of NICD. This zone around $r=0.9$ has been found interesting experimentally as well, where we observe a spike in HNF4a right before the edge (Fig 1B) and actin staining as well (Supplementary figure 1). Further, a similar force system modelled by theoretical physicists (<https://www.biorxiv.org/content/10.1101/2021.04.30.442205v1.full.pdf>) found that the cells right before the edge experience the most contraction from both the direction. This is an interesting finding, and we are exploring this for future studies, specifically on how this leads to higher HNF4a expression.

10. Authors should carefully review all figures to make sure panels haven't overlapped and placement is as intended. For example, the legend of Fig 6G and "1000" on Fig 6M y-axis.

Response: This has been fixed and checked for across all the figures, thank you for bringing it to our attention.

11. Line 664: "0.2 N NaOH" -> 0.2 M NaOH?

Response: This change has been incorporated in the main text, thank you for bringing it to our attention.

12. Line 757: biological

Response: This typo has been corrected in the main text, thank you for bringing it to our attention.

Reviewer #3 (Remarks to the Author):

The authors have tried to identify the role of Notch signaling and biomechanical cues during liver differentiation using both microarray and modelling approaches.

The claims made by the authors are well supported by both the experimental and model outcomes.

I recommend this work for publication without further corrections.

Response: Thank you for publication recommendation.

REVIEWERS' COMMENTS:

Reviewer #1 (Remarks to the Author):

I really appreciate the authors' efforts to address all the points I raised; I believe the paper has been significantly improved, and that it is now ready for publication.

Reviewer #2 (Remarks to the Author):

The authors have made good efforts to address the referee's concerns, and most are resolved now. The referee appreciates the significant additional work that has gone into this revision work.

I would like to make a couple more minor suggestions to help with final polishing:

-Authors have added black dashed lines in some figure plots at $R=0.95$ to indicate the interior:exterior boundary; it would be helpful to explain this is what this indicates in figure captions (or on the x-axis of the plots) as appropriate.

-Authors may want to add reference to the stability analysis added in SI Figure 8 somewhere in the main text, as this analysis adds significant credibility to their model.

I recommend this paper for publication with these minor corrections.

We thank the reviewers for their thorough review and well organized, helpful comments. Please find our point-by-point responses below which also indicate the relevant revisions made to the manuscript in response. In the revised manuscript provided, changes are **highlighted in blue font**. Small changes which correct grammatical errors, or improve readability, have also been made throughout the manuscript and may not be highlighted in blue. Changes made to content and to directly address reviewer comments have been highlighted.

Reviewer #1 (Remarks to the Author):

I really appreciate the authors' efforts to address all the points I raised; I believe the paper has been significantly improved, and that it is now ready for publication.

Response: Thank You for the recommendation.

Reviewer #2 (Remarks to the Author):

The authors have made good efforts to address the referee's concerns, and most are resolved now. The referee appreciates the significant additional work that has gone into this revision work.

I would like to make a couple more minor suggestions to help with final polishing:

-Authors have added black dashed lines in some figure plots at $R=0.95$ to indicate the interior:exterior boundary; it would be helpful to explain this is what this indicates in figure captions (or on the x-axis of the plots) as appropriate.

Response: Thank you for this suggestion. I The reviewer is correct in pointing out that the $R=0.95$ line was not explained. The label has been added to figure legends now, in the description of the line plots in every figure.

-Authors may want to add reference to the stability analysis added in SI Figure 8 somewhere in the main text, as this analysis adds significant credibility to their model.

Response: Thank you for this input. We have now referenced the SI Figure 8 in the main text in lines- 268-269, 304-305, 331-333, 360-361, corresponding to every perturbation.

I recommend this paper for publication with these minor corrections.

Response: Thank You for the recommendation.